# Climate reverses directionality in the richness–abundance relationship across the World's main forest biomes

Jaime Madrigal-González ◉ et al.#

More tree species can increase the carbon storage capacity of forests (here referred to as the more species hypothesis) through increased tree productivity and tree abundance resulting from complementarity, but they can also be the consequence of increased tree abundance through increased available energy (more individuals hypothesis). To test these two contrasting hypotheses, we analyse the most plausible pathways in the richness-abundance relationship and its stability along global climatic gradients. We show that positive effect of species richness on tree abundance only prevails in eight of the twenty-three forest regions considered in this study. In the other forest regions, any benefit from having more species is just as likely (9 regions) or even less likely (6 regions) than the effects of having more individuals. We demonstrate that diversity effects prevail in the most productive environments, and abundance effects become dominant towards the most limiting conditions. These findings can contribute to refining cost-effective mitigation strategies based on fostering carbon storage through increased tree diversity. Specifically, in less productive environments, mitigation measures should promote abundance of locally adapted and stress tolerant tree species instead of increasing species richness.

---

#A list of authors and their affiliations appears at the end of the paper.

More tree species can increase tree abundance through facilitation[1] and niche partitioning[2], and thereby enhance the forests' capacity to store carbon[3]. An increasing body of evidence points to close relationships between tree diversity and forest functions and services in tropical[4], temperate[5,6] and boreal biomes[7]. Maintaining and enriching tree assemblages could thus help mitigating climate warming via denser tree packaging and more efficient resource use by facilitation and/or niche partitioning[8]. Such a strategy will represent the positive effects of species richness on abundance through the existence of spatial segregation dynamics fuelled by intraspecific competition on sessile organisms' populations[2]. Secondary succession models provide evidence that a more efficient space filling and aboveground biomass accretion is achieved when species with different degrees of tolerance to shading coexist[9]. Growth–trait inference has recently been found to be contingent on tree species richness, with stronger influence of traits on growth in cases where tree diversity is higher[10]. However, recent evidence puts such a decisive role of species richness on forest carbon storage and other forest functions into question[11–14].

By contrast, higher available energy can promote species richness and tree biomass storage by promoting abundance[15]. This idea, framed as the more individuals hypothesis, assumes that the number of species is solely a probabilistic product of abundance in the sense that viability of natural populations is contingent on the number of available individuals[16]. This hypothesis has been supported in observational and experimental tests[15,17,18], and provides an almost mirror image of causal pathways for the relationship between species richness and abundance. Nonetheless, it seems that only the soft formulation of the hypothesis (which considers not only energy, but also environmental stochasticity driving richness) is plausible since species richness can indeed explain abundance patterns in some cases[19]. Any attribution of richness or abundance as the cause and/or the consequence of one another therefore remains a challenge. Yet, setting this

causal relation is a prerequisite to understand the likely mechanisms underlying the correlational evidence on the richness–biomass storage relationship available in forest ecosystems. Similarly, it remains unclear whether such a causal picture is idiosyncratic or would depend on environmental conditions[20]. In a context of ongoing climate warming[21], improved understanding of causality in the richness–abundance relationship across forest biomes could critically complement the suite of nature-based mitigation solutions and their effectiveness worldwide.

To this end, we explored the causal direction in the richness–abundance relationship using a dataset comprising more than 3000 forest plots distributed in 23 forest regions across the five forested continents (Fig. 1) for a total of more than 84,000 individual trees (see Supplementary Information S1, Table S1 for more details). All plots used here are located in natural, unmanaged forests within protected areas, such as to avoid, as much as possible, anthropogenic influence on abundance and tree species richness. We used structural equation models (SEMs)[22] to test whether species richness can be expressed as a cause (more species hypothesis) or as a consequence (more individuals hypothesis) of tree abundance in each forest region independently. Importantly, trees form size-structured assemblages where abundance is strongly constrained by tree size following the Yoda's law paradigm[23], stating that standing biomass increments, related to tree development, are compensated by self-thinning dynamics associated to space filling and the ensuing resource shortage[24,25]. This negative relationship between tree size and abundance is critical for a proper evaluation of the richness–abundance relationship. We therefore included a causal pathway from mean tree size to abundance in our theoretical SEM framework to account for variations in abundance associated to size dynamics (Fig. 2). The Bayesian information criterion (BIC) and the C statistic information criterion (hereafter CIC)[26] were used to establish the prevalence of either the more species or more individuals hypotheses in each of the 23 forest

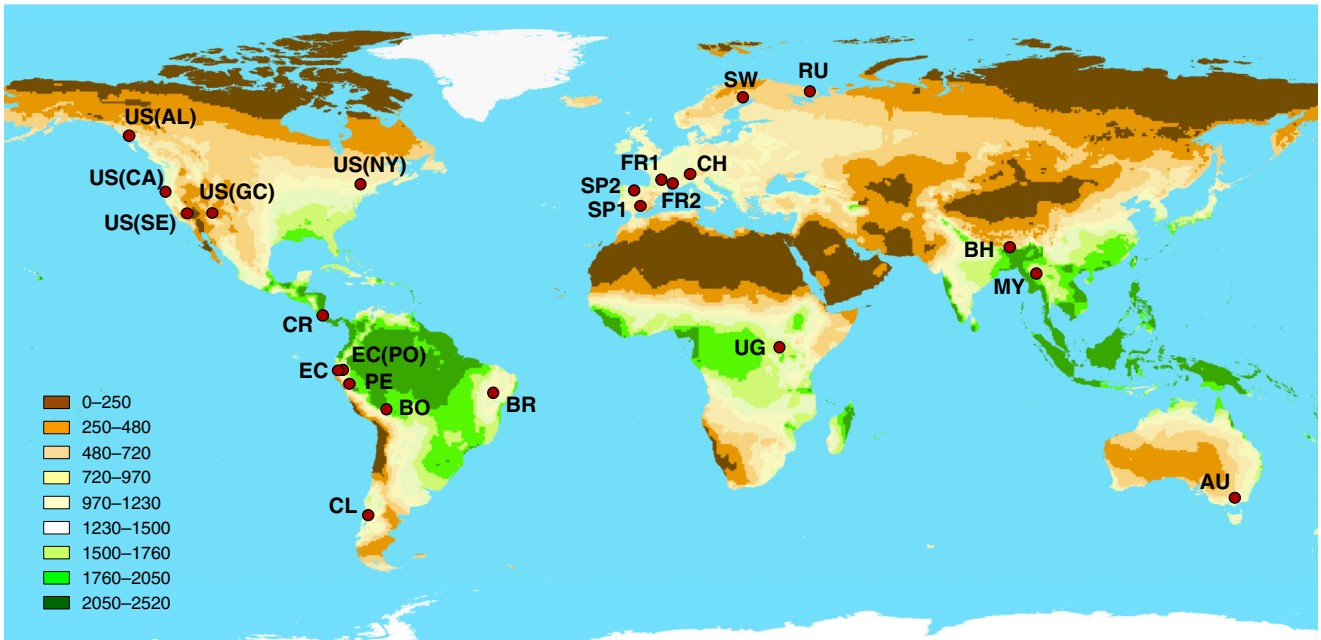

**Fig. 1 Geographical distribution of the forest regions studied.** The colour gradient (see legend) represents the climatological NPP (FAO's NPP index expressed in gDM m$^2$ yr$^{-1}$). Legend for acronyms: US(AL)—Alaska (US), US(CA)—Northern California (US), US(NY)—New York State (US), US(SE)—Sequoia National Park (US), US(GC)—Great Canyon (US), CR (Costa Rica), EC—Ecuador dry forest, EC(PO)—Podocarpus National Park (Ecuador wet), PE—Peru, Bo—Bolivia, BR—Brazil, CL—Chile, SP1—Spain (Sierra Nevada National Park), SP2—Spain (Fuentes Carrionas Natural Park), FR1—France (Cévennes National Park), FR2—France (Mercantour National Park), CH—Switzerland, SW—Sweden, RU—Russia, UG—Uganda, BH—Bhutan, MY—Myanmar, AU—Australia.

regions considered (see Supplementary Information S2 for comparison between BIC and CIC).

## Results

A richness–abundance relationship was found in 20 out of the 23 studied forest regions, whereas support to this relation was weak in the remaining three forest regions, with richness–abundance models showing no differences with a null model ($\Delta$BIC < 2). The more species hypothesis prevailed in eight forest regions (Table 1). In the other forest regions, the more species hypothesis was just as probable (nine regions with $\Delta$BIC < 2) or even less likely than the more individuals hypothesis (six forest regions). Thus, unlike commonly assumed[3], we evidence that the positive effects of species richness do not hold globally. Whereas such a lack of generality has been reported recently, the potential underlying causes remain

elusive[14]. The $\Delta$BIC indicates a prevalence of the more species hypothesis towards lower latitudes (Fig. 3), mostly in rainy tropical forests of Central (Costa Rica) and South America (Bolivia, Ecuador and Peru), and Africa (Uganda). Importantly, sensitivity analyses to (1) data harmonization (see Supplementary Information S3), and (2) the environmental factors used in the SEMs (see Supplementary Information S4) strongly supported the mentioned pattern. In productive forests, climatic conditions that remain stable throughout the year, could foster trait evolution towards complementary forms of resource use[27], in line with recorded increases in stand diversity towards the most productive and climatically stable environments on Earth[28]. Thus, our findings evidence the potentially decisive role of climate as a driving mechanism underlying the causal richness–abundance relationship.

To formally evaluate under which conditions one of the contrasting causal hypotheses prevails over the other at the global scale, we analysed the relation between the $\Delta$BIC (indicative of prevailing more species hypothesis when negative and more individuals hypothesis when positive) and the FAO's (region averaged) climatological net primary productivity (NPP) index using a SEM (similar results using NDVI instead of NPP were yielded; Supplementary Information S5). We obtained a negative relationship between the $\Delta$BIC (relative support to both hypotheses) and the climatological NPP (coefficient NPP = −0.46, $P$-value < 0.05, $R^2 = 0.50$), and showed that artefacts related to sampling protocols (differing number of plots; see Supplementary Information S6), latitude or species richness gradients can be excluded (Fig. 4). Otherwise, the outputs of the SEM (Fisher's $C = 7.382$, $P$-value = 0.496) suggest that a potential direct path between latitude and the $\Delta$BIC is negligible, and thus that latitudinal gradients other than climatic variability could be discarded. Therefore, for as long as temperature, precipitation, or the combination of both will limit NPP, mechanisms underpinning the more species hypothesis (such as complementarity) will have less influence than the climatic filtering has on functional strategies and tree abundance. Moreover, a positive feedback between both types of mechanisms might be plausible in the sense that in environmental contexts where more energy supports

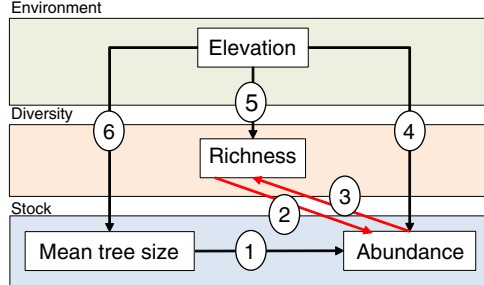

**Fig. 2 Causality in the abundance–diversity relationship.** Theoretical model arranged as a path diagram in which the arrows indicate the direction of potential causal effects. Path (1) represents Yoda's law in the sense that a negative relationship is expected between the size of trees and abundance per unit area. Paths (4), (5), and (6) represent gradients in abundance, species richness and mean tree size (i.e. mean diameter at breast height) along the elevation gradients. Paths (2) and (3) are nonrecursive paths indicative of more species and more individuals hypotheses, respectively. Comparison of models including one or the other path gives support to one or the other hypotheses, respectively.

---

**Table 1 Diversity effects were only supported in six forest regions.**

| Country | Region | $n$ | BIC_MSH | BIC_MIH | BIC_NULL | $\Delta$BIC_MSH-MIH | $\Delta$BIC_supp-NULL | Fisher's $C$ |
|---|---|---|---|---|---|---|---|---|
| Russia | Kola | 28 | 38.779 | 38.65 | 35.729 | 0.129 | 2.921[a] | 2.407 |
| US | Sequoya National Park | 132 | 59.903 | 62.438 | 94.089 | −2.535 | −34.186 | 6.192* |
| US | Great Canyon National Park | 229 | 94.24 | 83.382 | 161.567 | 10.858 | −78.185 | 7.31* |
| Sweeden | Northern Sweeden | 101 | 61.954 | 51.06 | 127.12 | 10.894 | −76.06 | 0.294 |
| Spain | Sierra Nevada National Park | 56 | 46.133 | 47.261 | 57.43 | −1.128 | −11.297 | 1.854 |
| Switzerland | Alps | 234 | 80.669 | 77.006 | 75.72 | 3.663 | 1.286[a] | 0.507 |
| Bhutan | Toepisa | 160 | 75.803 | 76.02 | 177.499 | −0.217 | −101.696 | 4.751 |
| US | Alaska | 491 | 96.893 | 88.159 | 248.759 | 8.734 | −160.6 | 1.409 |
| US | New York | 197 | 75.841 | 75.445 | 101.421 | 0.396 | −25.976 | 1.48 |
| Brazil | Bahia | 106 | 61.503 | 51.564 | 66.877 | 9.939 | −15.313 | 0.266 |
| Ecuador | Western Ecuador | 48 | 44.902 | 43.462 | 68.355 | 1.44 | −24.893 | 0.879 |
| Australia | Victoria | 44 | 46.95 | 42.955 | 46.344 | 3.995 | −3.389 | 1.329 |
| France | Mercantour National Park | 61 | 45.302 | 51.407 | 56.684 | −6.105 | −11.382 | 0.082 |
| Chile | Northern Patagonia | 109 | 71.179 | 71.031 | 118.788 | 0.148 | −47.757 | 5.352 |
| France | Cévennes National Park | 98 | 51.299 | 51.419 | 47.123 | −0.12 | 4.176[a] | 1.273 |
| Spain | Fuentes Carrionas Natural Park | 117 | 55.627 | 60.421 | 57.464 | −4.794 | −1.837 | 3.243 |
| US | Klamath Forest | 74 | 48.697 | 47.473 | 89.029 | 1.224 | −41.556 | 0.128 |
| Ecuador | Podocarpus National Park | 30 | 37.992 | 42.404 | 43.511 | −4.412 | −5.519 | 0.579 |
| Peru | Río Abiseo National Park | 30 | 40.796 | 50.038 | 49.742 | −9.242 | −8.946 | 3.383 |
| Myanmar | Wetphuyay | 62 | 59.836 | 58.427 | 103.986 | 1.409 | −45.559 | 0.647 |
| Uganda | National Park | 622 | 106.037 | 116.223 | 663.936 | −10.186 | −557.899 | 5.506 |
| Bolivia | Madidi National Park | 44 | 54.454 | 71.85 | 76.687 | −17.396 | −22.233 | 1.475 |
| Costa Rica | Costa Rica | 96 | 69.109 | 72.112 | 120.279 | −3.003 | −51.17 | 5.208 |

Model selection using the Bayesian Information Criterion (BIC). Fisher's $C$ is the statistic used to test for the existence of independent claims (not accounted paths) in the hypothesized model[36].
MSH more species hypothesis, MIH more individuals hypothesis.
*P-values < 0.05.
[a]Indicates the existence of missing paths that were unaccounted in the initial model.

more individuals, and this allows for more species (to be sampled from the regional pool), increased niche partitioning will mean then that even more individuals can persist. Future research should thus focus on unveiling such potential positive feedbacks between more energy and complementarity driving ecosystem functioning in forests and other ecosystems worldwide.

Our results question recent statements on biodiversity–ecosystem functioning using a unidirectional interpretation of the diversity–productivity relationship and, at a global level, found biodiversity to be critical for carbon sequestration[3]. Instead, we evidence that mechanisms underpinning this positive relation may differ quite clearly between forest biomes and that it can even reverse as a function of climate[20,29]. Previous findings in north America support the idea that opposing causal paths between species richness and productivity are plausible, and that

the intensity of such influences varies among biogeographical regions[7]. Our formulation, nonetheless, explicitly introduces abundance as a major component in this theoretical framework, thus allowing for the inclusion of the other way around between tree stocks and species richness. Noteworthy, the hypothesis of our study relies on the notion that species richness (#species) can efficiently summarize species diversity at the regional level. Thus, future investigation should strive to disentangle the roles of diversity dimensions other than species richness alone, namely functional and phylogenetic diversity. Such an approach would, however, represent a major challenge because it would neither be easy nor intuitive to define how more individuals can determine species combinations or simply affect functional or phylogenetic diversity. Further efforts should thus be paid to these crossroads as a potential way towards a new paradigm of thinking about the meaning of diversity–abundance relationships in natural communities and on how they can help refine the most classical biodiversity–ecosystem functioning paradigm.

The use of species mixing as an efficient low-cost strategy to enlarge forest carbon stocks will have to take account of the key limiting role that climate may have on species assembly and diversity[30]. As such, niche partitioning will be more likely in highly productive environments (in climatic terms) in which species' functional traits may have long evolved following prevailing pressures fuelled from co-occurring species[27,31]. In less productive environments, efforts to maximize carbon storage should be oriented towards maintaining abundance of locally adapted and stress-tolerant species through maximization of productivity. In any case, strategies should be considered in an adaptive framework, because climate change is likely to alter environmental conditions drastically in the future[32], and thereby also change the causal direction in the diversity–abundance relationship across biomes.

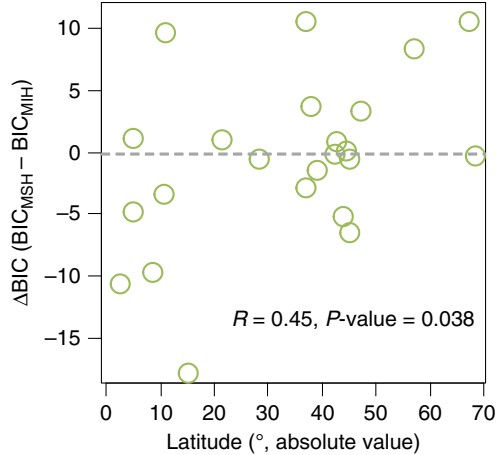

**Fig. 3 The more species hypothesis is more likely towards low latitudes.** ΔBIC (BIC$_{MSH}$–BIC$_{MIH}$) is negatively correlated to Latitude expressed in absolute values ($R = 0.45$, $P$-value = 0.038)). The dashed line represents the threshold above which the more individuals hypothesis (MIH) is more likely than the more species hypothesis (MSH); the opposite is true below the dashed line.

## Methods

**Forest inventory information.** We considered a total of 23 forest regions across the five forested continents on Earth. Of the 23 regions, five are located in North America (US), one in Central America (Costa Rica), six in South America (Ecuador (2), Brazil, Bolivia, Perú, and Chile), one in Africa (Uganda), one in Oceania

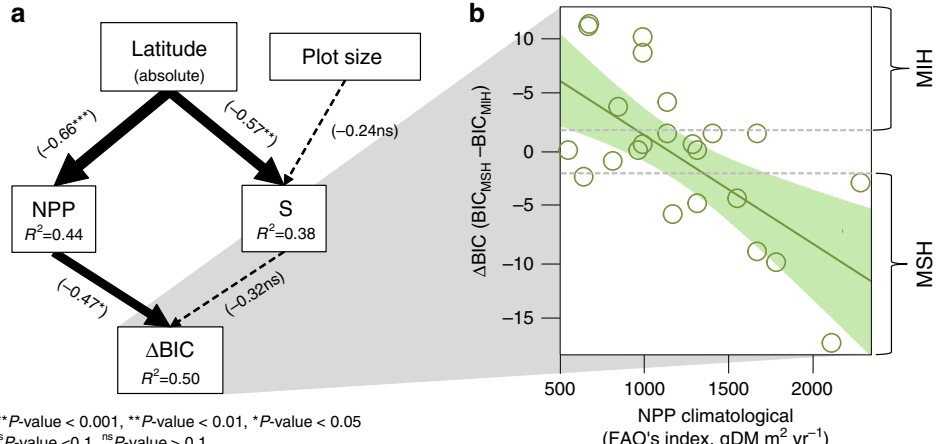

**Fig. 4 Climate controls on causality of the richness-abundance relationship.** The more species hypothesis is more likely in highly productive forest regions (climatologically speaking). **a** Graphical representation of the SEM outputs in which boxes represent the variables involved, arrows are illustrative of the causal paths, values in brackets denote the standardized coefficients (see legend of asterisks for $P$-value interpretation), and $R^2$ are the determination coefficients for the different regression models considered in the SEM. Solid arrows are indicative of significant pathways whereas dashed ones imply no significant relationships. **b** ΔBIC as linear function of NPP (FAO's climatological index). The shaded area is illustrative of the 95% confidence interval around the marginal mean (solid line in green). Dashed lines represent 2 units of ΔBIC above (+2) and below (−2) 0, which is indicative of equivalent support for both hypotheses. According to the state of the art in model selection using information criteria, we established a band between +2 and −2 ΔBIC units in which supports for one or the other hypotheses are equivalent (more species (MSH) and more individuals (MIH)). NPP net primary productivity; S species richness, BIC Bayesian information criterion.

(Australia), three in Asia (eastern Russia, Bhutan and Myanmar) and six in Europe (Sweden, Switzerland, France (2) and Spain (2)). From each forest region, we used forest inventories from the corresponding National Forest Inventories (US, Costa Rica, Chile, Spain, France, Switzerland, Sweden, Bhutan and Myanmar) or forest inventories obtained for clearly defined research purposes (Ecuador, Bolivia, Perú, Brazil, Uganda and Russia) with comparable protocols and clear sampling design. In all cases, circular or rectangular sampling plots with identical sizes within each forest region (although variable among the different forest regions), were distributed in extensive forested areas covering altitudinal gradients. Sampling plots in general were distributed following systematic protocols. A brief description of the datasets used in this study can be found in Table S1 and more detailed information can be found in the literature cited therein and in the supplementary material (Supplementary Information S7).

**Field measurements and biotic data**. Every tree with a diameter at breast height (DBH) exceeding 5–10 cm (depending on the forest inventory) was measured and identified at the species level, although a variable number of specimens in Bhutan, Myanmar, Ecuador and Peru were identified as morphospecies. Species richness was assessed as the number of species per plot, abundance as the number of standing tree individuals per plot and mean tree size as the average DBH per plot. Elevation and geographical coordinates were obtained in every plot using global positioning systems. The climatological NPP index[33,34] and maps of Köppen climate classes including updated temperature and precipitation records (CRU and GPCP VASClimO) were retrieved from: www.fao.org/nr/climpag/globgrids/ npp_en.asp[35]. This NPP index is assessed as a non-linear combination of temperature and precipitation following the equations in the Miami model[35]:

$$NPP_T = 3000(1 + \exp(1.315 - 0.119 \times T))^{-1}, \qquad (1)$$

$$NPP_P = 3000(1 - \exp(-0.000664 \times P)), \qquad (2)$$

$$NPP = \min(NPP_T, NPP_P), \qquad (3)$$

where $NPP_T$ and $NPP_P$ represent the components of primary productivity associated to temperature (T) and precipitation (P), respectively. NPP will increase with rising temperature and precipitation up to a saturation of 3000 gDM/m²/year (with DM standing for dry matter). Interestingly, this index assumes that both temperature and precipitation are limiting factors of the NPP.

**Statistical analyses**. To explore which of the two competing hypotheses is supported in each of the 23 forest regions considered, we analysed the direction of the richness–abundance relationship by applying SEMs following the theoretical framework defined in Fig. 2. The two hypotheses differ in the causal direction of the richness–abundance relationship (direction of the arrow linking species richness and abundance). The two candidate models share three equations: (1) abundance as a linear function of mean tree size and elevation, (2) richness as a linear function of elevation, and (3) mean tree size as a function of elevation. On the contrary, the SEM testing the more species hypothesis (MSH) included (4) abundance as a function of richness, whereas the SEM testing the more individuals hypothesis (MIH) included (5) richness as a function of abundance. Elevation was included as the main environmental gradient within each forest region to account for the influence of environmental variability on diversity, mean tree size and abundance. Elevation has been largely recognized a good composite variable of temperature and precipitation variability, but also climatic extremes[36], atmospheric pressure and edaphic conditions[37] in forests. On the other hand, elevation has been commonly used as an integrative environmental proxy explaining gradients of species richness, tree density and forest dynamics at local to regional scales[38–41]. We tested the roles of environmental variables other than elevation in a sensitivity analysis, namely elevation (second-order polynomial), climatological net primary productivity (NPP, Miami model), cation exchange capacity (CEC) and normalized difference vegetation index (NDVI). Results of these analyses can be found in the supplementary material (Supplementary Information S4). Both richness and abundance per plot, as discrete variables, were log-transformed to meet the normal distribution of residuals and homoscedasticity. Besides, the log-transformation of abundance allowed for the linearization of the relationship abundance-mean tree size, which is known to adopt negative exponential forms[24]. Inside the SEM models, we applied regression analyses using the function *lm* of the package *stats* in R[3,42]. In cases where sampling design imposed spatial dependencies among sampling plots we used a mixed-effect model with random structures (random intercept models) adapted to plot spatial clustering using the function *lme* of the package *nlme* in R[43]. Spatial clustering is characteristic for instance in the US National Forest Inventory, where sampling plots are distributed forming tracks (1–4 sampling plots) across the territory. Similarly, the NFIs from Bhutan, Myanmar and Chile are stablished forming grids where every node represents a track, and each track contains a maximum of four plots. In the case of Bolivia, data are clustered by localities. Sampling plots in Costa Rica are clustered in large regions representative of different forest types (mature forest, secondary forest, Manglar, Yolillal forest and deciduous forest). Finally, a transect scheme in which sampling plots are distributed linearly across the territory was applied in Uganda.

Model comparison was realized using the BIC. Specifically, we assessed ΔBIC as

$$\Delta BIC = BIC_{MSH} - BIC_{MIH}, \qquad (4)$$

where $BIC_{MSH}$ is the BIC associated to the MSH, and $BIC_{MIH}$ is the BIC associated to the MIH. This values of ΔBIC < −2 were indicated a prevailing role of species richness as a predictor of abundance. On the contrary, values > +2 in the ΔBIC support a prevailing role of abundance as a predictor of species richness. Between −2 and +2, both causal hypotheses are assumed to be likely equivalent and thus, both types of mechanisms can have a role in driving species assembly at the plot level. It is noteworthy that, and even if one of the hypotheses is prevalent over the other, this does not imply a lack of mechanisms associated to the other hypothesis, but a comparatively less relevant one. Similarly, that one model prevails over the other does not necessarily implies a causal effect. Instead, a null model (i.e. no effect of species richness on abundance or vice-versa) might perform better than the prevailing mechanistic model. Thus, to test the effect of species richness on abundance (under the more species hypothesis) and abundance on species richness (under the more individuals hypothesis), we compared the prevailing model (if any) with a null model using BIC. Goodness-of-fit of the best-supported model was analysed using Fisher's C statistic under the null hypothesis that no independent claims exist that improve the current SEM structure. SEM models were analysed using the *psem* function in the *piecewiseSEM* package in R[26].

To analyse whether climate controls the prevalence of one hypothesis over the other at the global scale we used a SEM with *piecewiseSEM* package in R. We modelled the ΔBIC as a function of the climatological NPP but also included species richness as a potential confounding effect of latitude, and plot size differences among regions to control for the species richness variability as an artefact associated with the different sampling protocols regionally. We thus evaluated the role of species richness as a potential direct predictor of the ΔBIC in the presence of NPP. In doing so, we can evaluate whether the observed pattern in ΔBIC is just a consequence of a confounding relationship between species richness and NPP along the latitudinal gradient or determined by a net climatological influence. As a sensitivity analysis for the use of NPP as a global-scale environmental variable we used NDVI (1 km resolution) instead of the climatological NPP. Results suggested that, even if the pattern holds for NDVI consistently with NPP, the climatological NPP was a better candidate for the model ($BIC_{NPP} = 41.87$, $BIC_{NDVI} = 63.73$; Fig. S5). Additionally, we tested the support for the missing paths (latitude on ΔBIC) in the SEM using Fisher's C statistic. Specifically, missing paths will be negligible when the P-value associated to the Fishers C statistic is larger than 0.05.

**Reporting summary**. Further information on research design is available in the Nature Research Reporting Summary linked to this article.

## Data availability
Part of the data that support the findings of this study are available from [INFOR Chile, lfi.ch Switzerland] but restrictions apply to the availability of these data, which were used under license for the current study, and so are not publicly available. The rest of the data is publicly available at https://figshare.com/account/home (https://doi.org/10.6084/m9. figshare.13072211).

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

## Acknowledgements

A.E. was supported by REMEDINAL TE-CM (S2018/EMT-4338). L.C. and M.J.M. were supported through two grants from the Spanish Ministry of Economy and Competitiveness (CGL2013-45634-P, CGL2016-75414-P). M.A.Z. was supported by grant RTI2018-096884-B-C32 (MICINN, Spain). C.A. was supported by the Victorian DELWP iFER (Integrated Forest Ecosystem Research) programme. A.H. was supported by the University of Alcalá (Own Research Programme 2019 Postdoctoral Grant) and Basque Country Government funding support to FisioClimaCO2 (IT1022-16) research group. We thank the MITECO and MAPA (Spain) for granting access to the Spanish Forest Inventory Data.

## Author contributions

J.M.-G. and M.S. conceived the main ideas; J.M.-G., J.C. and J.B.C. analised the data; J.M.-G., J.C., A.E., L.C. and M.S. generated the first draft of the manuscript; L.C., M.R., P.R.-B., C.A., R.S., A.J.P., S.D., C.I.E., O.T., M.M., L.P., M.J.M., M.A., A.Q.-R., M.V.-A., E.G., Y.T., provided data; J.M.-G., J.C., J.B.C., A.E., L.C., M.R., P.R.-B., A.H., C.A., R.S., A.J.P., S.D., C.I.E., O.T., M.M., L.P., J.L.-S., M.J.M., M.A., M.A.Z., A.Q.-R., M.V.-A., E.G., Y.T., M.S. contributed to the writing of the last version of the manuscript.

## Competing interests

The authors declare no competing interests

## Additional information

Jaime Madrigal-González [1✉], Joaquín Calatayud[2,3], Juan A. Ballesteros-Cánovas[1,4], Adrián Escudero[3], Luis Cayuela [3], Marta Rueda[5,6], Paloma Ruiz-Benito[3,7], Asier Herrero[7], Cristina Aponte [8,9], Rodrigo Sagardia[10], Andrew J. Plumptre [11], Sylvain Dupire [12], Carlos I. Espinosa [13], Olga Tutubalina [14], Moe Myint[1], Luciano Pataro[15], Jerome López-Sáez[1], Manuel J. Macía[15,16], Meinrad Abegg [17], Miguel A. Zavala[7,18], Adolfo Quesada-Román[1,19], Mauricio Vega-Araya[20], Elena Golubeva [14], Yuliya Timokhina[14] & Markus Stoffel[1,4,21]

[1]Climate Change Impacts and Risks in the Anthropocene (C-CIA), Institute for Environmental Sciences (ISE), University of Geneva, 66 Boulevard Carl Vogt, CH-1205 Geneva, Switzerland. [2]Integrated Science Lab, Department of Physics, Umeå University, 901 87 Umeå, Sweden. [3]Departamento de Biología y Geología, Física y Química inorgánica. ESCET, Universidad Rey Juan Carlos, C/Tulipán s/n, Móstoles, C.P. 28933 Madrid, Spain. [4]Department of Earth Sciences, University of Geneva, 13 rue des Maraîchers, CH-1205 Geneva, Switzerland. [5]Department of Conservation Biology, Estación Biológica de Doñana CSIC, Sevilla, Spain. [6]Departamento de Biología Vegetal y Ecología, Universidad de Sevilla, C/ Profesor García González s/n, 41012 Sevilla, Spain. [7]Forest Ecology and Restoration, Departamento de Ciencias de la Vida, Universidad de Alcalá, ctra. Madrid-Barcelona, km 33.4, 28805 Alcalá de Henares, Spain. [8]School of Ecosystem and Forest Sciences, The University of Melbourne, 500 Yarra Boulevard, Richmond, VIC 3121, Australia. [9]National Institute for Research and Development in Forestry "Marin Dracea", 128 Blvd. Eroilor, Voluntari 077190 Ilfov, Romania. [10]Instituto Forestal de Chile, Sucre 2397, Ñuñoa, Santiago de Chile, Chile. [11]KBA Secretariat for KBA Partnership, Cambridge, UK. [12]Université Grenoble Alpes, Inrae, LESSEM, 38000 Grenoble, France. [13]EcoSs_Lab, Departamento de Ciencias Biológicas, Universidad Técnica Particular de Loja, San Cayetano Alto, 110107 Loja, Ecuador. [14]Faculty of Geography, Lomonosov Moscow State University, Moscow, Russia. [15]Departamento de Biología (Botánica), Facultad de Ciencias, Universidad Autónoma de Madrid, calle Darwin 2, Madrid, Spain. [16]Centro de Investigación en Biodiversidad y Cambio Global (CIBC-UAM), Universidad Autónoma de Madrid, Calle Darwin 2, ES–28049 Madrid, Spain. [17]Swiss Federal Institute for Forest, Snow and Landscape Research, WSL, Zürcherstrasse 111, 8903 Birmensdorf, Switzerland. [18]Instituto Franklin, Universidad de Alcalá, Calle Trinidad 1, 28801 Alcalá de Henares, Madrid, Spain. [19]Escuela de Geografía, Facultad de Ciencias Sociales, Universidad de Costa Rica, Ciudad de la Investigación, Montes de Oca 2060, San José, Costa Rica. [20]Instituto de Investigación y Servicios Forestales (INISEFOR), Universidad Nacional de Costa Rica, 86-3000 Heredia, Costa Rica. [21]Department F.-A. Forel for Environmental and Aquatic Sciences, University of Geneva, 66 Boulevard Carl Vogt, CH-1205 Geneva, Switzerland. ✉email: Jaime.MadrigalGonzalez@unige.ch

