## [Peer Review File · Nature Communications]

Reviewers' Comments:

Reviewer #1:

Remarks to the Author:

In this study, the authors aimed at revisiting the old debate about the direction of the relationship linking diversity and productivity in forest ecosystems, through the angle of the diversity-abundance relationship. To do so they rely on the analysis of a global dataset of forest plots, to 1) test the relative prevalence of two hypotheses (diversity -> abundance vs. abundance -> diversity) and 2) test whether the change of prevalence between the two hypotheses may change with productivity (NPP) at the regional scale.

Although this study does not include any test the effect of diversity and/or abundance on NPP, which could have been useful regarding the main aim of the paper, the comparison proposed here remains certainly interesting and timely. However, I think that this study suffers from many flaws and over-interpretation of the results, while the literature citations were not always very accurate, and key references missing.

Major concerns

First of all, the authors mention the « causality » issue overall the manuscript (notably in the title of the study). However, SEM remains a statistical tool. SEM may detect more sophisticated correlations between variables and help in determining a given variance-covariance structure among the variables, and I do not deny their usefulness. However, strictly talking about « causality » when using SEMs is an overstatement, and a lot of SEM experts I have talked with actually call for caution regarding this topic. For instance:

Bargozzi R. P. 2010 Structural equation models are modelling tools with many ambiguities: Comments acknowledging the need for caution and humility in their use. *Journal of Cons. Psy.*
<https://doi.org/10.1016/j.jcps.2010.03.001>

Second, my main concern deals with the mismatch between the hypotheses tested and the data used to do so. The Table S1 gives the number of plots per forest region. To me, there are too few plots per region to draw any conclusion. When dealing with diversity issues, you need many repetitions per level of diversity. In each « region » chosen, the environmental conditions greatly varies across plots. Although the range of diversity is not given at the scale of forest regions, one may easily suspect that it ranges from 1 to 10 in Austria, France, and Spain for instance. Then, analysing less than 100 plots for each region (country), even 35 for Austria, raises serious concerns.

This is also very surprising, because these NFI datasets include many more plots. I have some experience of the European NFI datasets, and I am surprised that the authors did not use more plots available. I understand that selecting plots without management legacy reduces the number of plots considered (see my comment on this point below), but this should have led to a much larger dataset still. The Spanish and French NFI include tens of thousands of plots. I also have to say that the references used in Table S1 are outdated for the NFI data I know the best, which may explains this limit.

The possible bias due to forest composition is also never mentioned or discussed. Forest plots with the same species richness may greatly vary in terms of functional composition, with consequences about tree abundance and species coexistence. In temperate/boreal forests (i.e. forest with less species per plot than in other biomes, on average), this effect is even a key one. This is actually what is illustrated by the Ref15 cited by the authors, although (as or even more) relevant studies on this issue are: Bohn, F. J., and A. Huth. 2017. The importance of forest structure to biodiversity-productivity relationships. *Royal Society Open Science* 4.

Maréchaux, I., and J. Chave. 2017. An individual-based forest model to jointly simulate carbon and tree diversity in Amazonia: description and applications. *Ecological Monographs* 87:632–664.

Morin, X., L. Fahse, M. Scherer-Lorenzen, and H. Bugmann. 2011. Tree species richness promotes productivity in temperate forests through strong complementarity between species. *Ecology Letters* 14:1211–9.

It is also key to know whether the species present in the plots with only one species are still present in the plots with n species. Otherwise, there is a bias and the monospecific stands cannot be fairly compared with the stands with more than 1 species.

Another major concern: the management issue is central in this kind of study, as management strongly influences species composition and tree abundance in forests. I found the justification about plot selection should be strengthened (see my comment above). Pristine forests do not exist any more in many places, eg. Spain or France. The precise methodology for plot selection is missing. At least the authors should have provided an averaged number of years without management for the plots of each forest region.

About SEMs, the AIC comparison is not the state-of-the-art of model comparison. Are the same conclusions arise when using other metrics, for instance χ^2 , RMSEA or BIC, which are more recommended in SEM. At least these other metrics should be provided in a supplementary material. See for instance:

Li-Chung Lin, Po-Hsien Huang & Li-Jen Weng. 2017. Selecting Path Models in SEM: A Comparison of Model Selection Criteria. *Structural Equation Modeling: A Multidisciplinary Journal*, 24: 855-869. doi.org/10.1080/10705511.2017.1363652

I have also a major concern regarding the environmental data used. In Fig. 2, why is elevation the only environmental predictor? In most temperate and boreal regions, climatic and soil conditions will not only vary with elevation, but also across space. And as far as it is described, the plots used for each region are spread all over the "region" (often a country). Moreover, incorporating latitude or longitude is not enough, true climatic variables should be used, especially when global climate datasets at high resolution are freely available (e.g. <http://chelsa-climate.org/>).

In the same vein, I have doubts about the NPP values used in Fig. 3 (ref #35). Why not having used more recent proxies, eg. derived from NDVI data, again freely available at global scale?

Other concerns

Several references were inaccurate in my opinion. For instance:

- Ref #1 does not deal with trees
- Ref #9 deals with deforestation, which is different than the effect of species loss in the context of the sentence (l. 28-30)
- Ref #13 is not strictly about boreal forests
- About Ref #15 & 16 (aimed at justifying space filling, eg. root segregation and canopy packing, although these terms are not mentioned): mentioning the relevance of canopy packing in ecosystem functioning with outputs from a model is a bit weak, a better reference could have been Jucker et al. (2015). Ref#16 is a bit controversial, as many works did not find any evidence for root segregation effects, eg. Meinen et al. 2009

Jucker, T., O. Bouriaud, and D. A. Coomes. 2015. Crown plasticity enables trees to optimize canopy packing in mixed-species forests. *Functional Ecology* 29.

Meinen C. et al. 2009. No evidence of spatial root system segregation and elevated fine root biomass

in multi-species temperate broad-leaved forests. *Trees*. 23:941-950

A relevant reference to introduce the topic could have been Waide et al. (1999):

Waide, R. B., M. R. Willig, C. F. Steiner, G. G. Mittelbach, L. Gough, S. I. Dodson, G. P. Juday, and R. Parmenter. 1999. The relationship between productivity and species richness. *Annual Review of Ecology and Systematics* 30:257–300.

Important information about the SEM results are missing to properly assess them. The authors should provide the coefficient values for each of their arrows, for each forest region and hypotheses tested, in supplementary material for instance.

Self-thinning in forests is not a linear relationship. Therefore the link between tree size and abundance cannot be linear. See for instance:

Sonja Vospernik, Hubert Sterba. Do competition-density rule and self-thinning rule agree? *Annals of Forest Science*, Springer Verlag/EDP Sciences, 2015, 72 (3), pp.379-390. [ff10.1007/s13595-014-0433-xf](https://doi.org/10.1007/s13595-014-0433-xf). [ffhal-01284181f](https://doi.org/10.1007/s13595-014-0433-xf)

Moreover, average tree size should be calculated with a quadratic mean, not arithmetic mean. See https://www.fs.fed.us/pnw/olympia/silv/publications/opt/436_CurtisMarshall2000.pdf

L. 93-94 : this is not really new, and I am surprised that the authors do not discuss the evidence for such a pattern shown in former studies (eg. Paquette & Messier 2011, cited in the study).

I. 109-111 : Again this is not a really novel finding (see. Paquette & Messier 2011, Jactel et al. 2018) who all show how climate controls the diversity-ecosystem functioning in forests), and we all agree that the Liang et al. study is partly biased because of an oversimplification of the pattern.

Jactel, H., E. S. Gritti, L. Drössler, D. I. Forrester, W. L. Mason, X. Morin, H. Pretzsch, and B. Castagneyrol. 2018. Positive biodiversity –productivity relationships in forests: climate matters. *Biology Letters* 14.

The authors mention that the plot data used follow the same kind of methodology across regions. This is not true. For instance in Europe, the French NFI data do not rely on permanent plots, while the Swiss and Spanish one do.

Futhermore, regarding the French NFI data, the authors do not mention how they dealt with the « simplified trees » present in the basic inventory. Indeed, a weight in these data should be calculated according to the size of the concentric plot in the French protocol (6m, 9m, 15m for small, medium, and large trees). This leads to weights of 14.14, 39.29, and 88.42 depending on the size of the tree. However, because some plots are in the border of the forest (therefore not having full size), or because of other subtleties in the protocol, the real weights of the trees differ from these values for 17% of them. And even more important: 30% of the plots contain trees for which the weight is wrong, and therefore the dendrometric values of 30% of the plot are wrong. Therefore the weights of the trees dataset need to be corrected using the French data, and this is not mentioned at all in the ms.

Table S1 : Species richness is known to vary with plot size. How this has been taken into account?

Reviewer #2:

Remarks to the Author:

This manuscript conducts interesting analyses and obtains intriguing results. The authors propose a framework to unify latitudinal gradients in tree diversity and productivity with general BEF theory by incorporating abundance-driven effects. They find diversity effects prevailing in productive (low

latitude) forests only, while in less productive forests abundance-driven effects overrule possible diversity effects.

I think the analytical approach (i.e. the core results shown in Figure 3) is novel and smart. If the results hold to further scrutiny, this manuscript has potential to become an important contribution to the field.

Nevertheless, I have several conceptual concerns and a few methodological questions.

Major concerns:

(1) In my opinion it is very naïve to assume that you can account for environmental variation by including elevation as a surrogate. No doubt, elevation is correlated with many environmental variables, but for a global analysis I highly doubt that elevation effects are consistent among regions and climates. What is missing are biologically relevant environmental variables related to productivity. Soil parameters come to my mind and would be appropriate, but there are likely other possibilities. As it stands, your analysis is missing an important dimension of tree growth (and thus productivity): soil fertility. It could easily be that your results are at least partly driven by differences in soil characteristics, considering that many of the highly weathered tropical soils are actually rather poor in nutrients, which might at least partly explain your latitudinal results. You should either add an analysis showcasing that elevation does, as you claim, account for environmental variation within and among forests regions, or add relevant additional data. To keep your current analytical framework (which I like) one could condense environmental variables to one or two principal components and use them as predictors in the path models.

(2) I know you want to sell your paper with global warming, but it is unclear to me how this feeds back into your analysis. Generally, I think you are overselling a lot. Your basic ecological findings are interesting enough. No need to construct an utter link to global warming, also when considering that temperature is just one of several components of NPP.

(3) The text is unclear in parts. Many concepts are mixed or not used in the appropriate context. This might potentially limit the accessibility of the manuscript for a general readership. I will point out a few problematic passages in the line-by-line comments below. For example, the 'diversity effects' hypothesis named by you is in the literature usually referred to as 'biodiversity-ecosystem functioning' or short 'BEF'.

(4) Actually, a part of your core result has been demonstrated before. Dormann et al. (<https://doi.org/10.1101/524363>) reanalyzed the dataset of Liang et al. (your reference 3), finding that the claimed general and positive relationship between tree diversity and productivity only holds for few ecoregions. This matches well with your findings.

Methodological questions:

(5) How were the data selected. Was there some sort of data filtering? This is currently obscure but very important information.

(6) How did you account for the differing forest inventory methods among forest regions? Whether you measure DBH for individuals exceeding 5 cm or 10 cm matters a lot for stem counts (i.e. abundance). This might have a large impact on the outcome of tests of the 'more individuals hypothesis'.

(7) As you are probably aware, the formulation of AICc depends on sample size (i.e. number of plots in your case). A difference in 2 AICc units means something different if you apply it to a dataset with 40 replicates or if you apply it to a dataset with 400 replicates. In the less replicated data, 2 units mean a relatively larger difference in model fit compared to the dataset with tenfold more replicates. According to your Table S1, number of plots varied by more than one order of magnitude. You should account for this in your analyses, if you want to compare AICc difference among forest regions. At least you have to demonstrate that the relationship in Figure 3 is not systematically influenced by differences in plot numbers. In this regard, just referring to the correlation coefficients in Figure S1 is

not sufficient.

Minor comments (line-by-line):

L 4: Packaging: This is too much jargon for the opening sentence.

L 6: See the Dormann et al. preprint referred to above for an opposing view.

L 9: You critically omit soil fertility from your considerations.

L 19ff: I can not see how your findings have the claimed 'major impacts' on climate change mitigations. Generally, you are overselling your results. The conclusions here are extremely vague and can not directly be deduced from your findings.

L 28: It is often claimed that forests are losing tree species. Yet, this is mostly due to forestry or maybe invasive species. As you have worked in unmanaged forests only, I am not sure that this strong statement is justified here.

L 34-35: What you frame as 'diversity effects' hypothesis has long been named as 'biodiversity-ecosystem functioning'.

L 40: It might be important to add that growth-trait inference depends on tree diversity, with usually stronger influences of traits on growth when tree diversity is larger. See the recent paper by Bongers et al (2020, Journal of Ecology 108: 256-266).

L 44: Be aware that the formulation of the 'more individuals hypothesis' uses energy as the primary currency. The problem I see here is that relative productivity (if we take it as surrogate for energy) changes in trees when trees grow (i.e. when abundances get lower but productivity at individual level gets larger). This might or might not be a violation of the principal assumptions of the MIH and it would be good if you develop this thought further.

L 61: How were the 21 regions/datasets selected? This is important to know.

L 72ff: This is a really smart analysis.

L 99: Delete 'very strong'. In general, you could tune down on the valuing language.

L 120: What is 'proper'? Unclear.

L 120ff: Again, I think this is too much overselling in regards to climate change.

L 121: The multifunctionality statement can also be made for productive sites. See e.g. Schuldt et al. (2018, Nature Communications 9: 2989) for a study from tropical forests.

L 138: 'comparable protocols and clear sampling design'. This is totally vague. We need much more detail in a supplementary file.

L 168: As explained above, I question whether elevation is sufficient for capturing between plot variation.

L 174: Please be more specific. For what models did you add random effects. What was the random effect structure? Random intercept or random slope models?

Figure 3: Please add the fitted mean to the plot.

Table S1: (1) Please unify the number of digits you give. For example, three digits for elevation measurements make no sense at all. Similar issues prevail for almost all columns. Make sure the displayed number of digits match to the accuracy of measurements.

Table S1: (2) Column 'NPP_median': Isn't it strange that BR and US(NY) have exactly the same NPP values? Same problem for other pairs (e.g. US(SE) vs. RU).

Table S1 (3) How were richness values intrapolated?

Reviewer #3:

Remarks to the Author:

The study aims to look at the relationship between richness and abundance in forest plot data, and to determine the casual direction of this relationship using SEMs. I think the question (what is the causality of the relationship) is interesting, and the dataset is impressive. However, there are several shortcomings that need to be addressed.

Main Points

One of the main issues is that the English needs a lot of improvement. This is not just a pedantic complaint, but I really struggled to understand a lot of what was written, particularly in the methods, which means it is hard to know exactly what was done. Some examples:

Lines 173 -175 – more information is needed about what this test is for and how it is implemented

203 – 206: I am not sure what this analysis is doing?

207 – 208: what does trusted missing paths mean?

There are other predictors that should be used in the SEMs. Plot size for one – I think this is a major variable to not be kept constant. It is well established that richness increases with area. Why not include plot size in the SEMs as an additional co-variate driving richness? You could also use mixed effects SEMs (which you briefly mention) and include plot inventory and/or region as random effects. I think these also need to be accounted for.

The fact that the minimum threshold of trees included varies between plots could also affect results. Why not run a sensitivity test and filter out all trees below 10cm (i.e. the highest minimum threshold across studies) DBH in all plots and re-run the analyses?

What happens with the goodness of fit test if Fisher's C is significant? Is the model excluded? Would it not make more sense to exclude it before comparing AICc? So as a first step test the goodness of fit of your two competing models to determine if they are acceptable, and if both pass this step, then compare?

It is not clear why you need to use elevation as a proxy and not climate variables themselves?

Other Points

I think it needs to be clearer early on how gaining a better understanding of causality in the relationship will help climate change mitigation. You start by talking about biodiversity crises and ecosystem functioning etc, but the link between these big issues and this particular relationship is not clear. In the final paragraphs at the end of the paper it becomes clearer, but perhaps stress this early on as well. An example from early on in the paper: you state "Maintaining and enriching tree assemblages could thus help mitigating climate change via more dense packaging and efficient resource use". But, based on what is written before this sentence, the link here is not clear.

Lines 43 – 46: Could be clearer that the More Individuals Hypothesis states that a greater amount of energy means more individuals which means more species. That first part of this (i.e. the energy part) is unclear in your description

48 – 49: But it has also been found not to hold in a large number of cases. The Storch et al paper you cite also distinguishes between the strict formulation and the weaker formulation of the hypothesis, which I think is worth mentioning. And they argue based on their theory that the strict formulation does not hold, but this is not mentioned.

Is diversity – abundance actually an accurate description? As diversity metrics usually incorporate a measure of abundance. Perhaps richness – abundance relationship is more accurate.

The wording around the use of AICc needs changing throughout the text. For example, lines 74-75, you mean AICc was used to compare models to test the prevalence ... And line 85, say something like

"Our results indicate". 99-100: I am not sure what delta AICc supporting diversity effect means?

Line 100 – add the coefficient for NPP as well as the R2

Reviewers' comments:

Reviewer #1 (Remarks to the Author):

In this study, the authors aimed at revisiting the old debate about the direction of the relationship linking diversity and productivity in forest ecosystems, through the angle of the diversity-abundance relationship. To do so they rely on the analysis of a global dataset of forest plots, to 1) test the relative prevalence of two hypotheses (diversity -> abundance vs. abundance -> diversity) and 2) test whether the change of prevalence between the two hypotheses may change with productivity (NPP) at the regional scale.

Although this study does not include any test the effect of diversity and/or abundance on NPP, which could have been useful regarding the main aim of the paper, the comparison proposed here remains certainly interesting and timely. However, I think that this study suffers from many flaws and over-interpretation of the results, while the literature citations were not always very accurate, and key references missing.

Thank you very much for the positive comments on our manuscript. We have done important changes and responded to every query to shed light on the key aspects demanding further clarification.

Major concerns

First of all, the authors mention the « causality » issue overall the manuscript (notably in the title of the study). However, SEM remains a statistical tool. SEM may detect more sophisticated correlations between variables and help in determining a given variance-covariance structure among the variables, and I do not deny their usefulness. However, strictly talking about « causality » when using SEMs is an overstatement, and a lot of SEM experts I have talked with actually call for caution regarding this topic. For instance:

Bargozzi R. P. 2010 Structural equation models are modelling tools with many ambiguities: Comments acknowledging the need for caution and humility in their use. Journal of Cons. Psy. <https://doi.org/10.1016/j.jcps.2010.03.001>

We agree that causality is hard to be tackled using correlational data only. Causality is however assumed in every regression model fitted using observational data, and so this uncertainty is implicit to every observational study in ecology and biogeography (and almost every scientific field using observational data). Importantly, structural equation models often allow to

improve our ability to deal with causality using observational data because potential confounding and indirect causal paths are considered within a structured frame of regression models (Shipley 2016). Along this line of reasoning, we even moved beyond the common use of SEM by considering that causality in the abundance-richness relationship can be evaluated bidirectionally (i.e. that the relationship, if existing, could be interpreted in either direction). Most of the empirical evidence in support to a biodiversity-ecosystem functioning relationship assumes a unique causal direction, and thus it is commonly accepted that biodiversity causes ecosystem function even if only observational evidence is provided (see most of the cases in the literature regarding forest ecosystems). It is important to remind that causality can be properly assigned by conducting two different approaches; first, an experimental control in which the different potential sources of causal variation are experimentally controlled; and, two, by including a statistical control. For instance, in the traditional SEM approach what we usually do is to compare the variance-covariance matrix for a set of observational data with the expected matrix if the hypothesized causal relationships are true. This is the reason why most authors have highlighted that SEM shouldn't be used in a descriptive framework but only with a clear hypothetical causal schedule.

That said, to meet the reviewer's view on the causality concept, we reworded the title using the term causal direction, which is more in line with the methodological perspective and less declarative in the (let's say) philosophical sense.

**Shipley, B (2016). Cause and correlation in Biology: a user's guide to Path Analysis, Structural Equations and Causal Inference with R. Cambridge University Press, Cambridge, UK
(<https://doi.org/10.1017/CBO9781139979573>)**

Second, my main concern deals with the mismatch between the hypotheses tested and the data used to do so. The Table S1 gives the number of plots per forest region. To me, there are too few plots per region to draw any conclusion. When dealing with diversity issues, you need many repetitions per level of diversity. In each « region » chosen, the environmental conditions greatly varies across plots. Although the range of diversity is not given at the scale of forest regions, one may easily suspect that it ranges from 1 to 10 in Austria, France, and Spain for instance. Then, analysing less than 100 plots for each region (country), even 35 for Austria, raises serious concerns.

This is also very surprising, because these NFI datasets include many more plots. I have some experience of the European NFI datasets, and I am surprised that the

authors did not use more plots available. I understand that selecting plots without management legacy reduces the number of plots considered (see my comment on this point below), but this should have led to a much larger dataset still. The Spanish and French NFI include tens of thousands of plots. I also have to say that the references used in Table S1 are outdated for the NFI data I know the best, which may explain this limit.

Thank you very much for taking this discussion up. We understand your worries on the data used, but we have to express our disagreement regarding the idea of the necessity of further data replication in our regression analyses. Replication of one predictor along the range of values of another predictor is necessary only if an interaction between them is considered in the model. In such a case, replicability allows for a proper evaluation on how the effect of a predictor on a response variable varies depending on the value of a second predictor. In cases where one is only considering main effects, one needs enough replicates to reliably estimate model parameters. Although is a kind of rule of thumb in some classical textbooks such as the "A primer of Ecological Statistics" (Gotelli and Ellison 2012), 10 replicates per parameter estimate are enough. In any case you can find numerous examples with fits of linear regression models with one predictor variable using less than 10 replicates (7-9). In the case of graph cyclic SEM statistics, the focus is on the regression models considered. Thus, the regression having the highest number of predictors will be the reference to consider as the minimum number of replicates. In our case, such a regression model contains three predictors, and so a sufficient number of replicates might be 30. Only in one of the datasets included in this work (Kola, Russia), this number was not met with a total of 28 replicates. Nonetheless, we left this dataset in the analyses by assuming that 28 replicates are still largely sufficient to fit the three-predictors regression model. In either case, low replicability is expected to decrease the statistical power of a test, i.e. the probability to reject the null hypothesis. Thus, our results are quite conservative as they are able to detect, despite the apparently low statistical power of one particular datasets, different relationships between response and predictor variables within the context of model comparison with information criteria.

Gotelli, N.J., Ellison, A.M. (2012). *A primer of Ecological Statistics*. Oxford University Press.

The possible bias due to forest composition is also never mentioned or discussed. Forest plots with the same species richness may greatly vary in terms of functional composition, with consequences about tree abundance and species coexistence. In

temperate/boreal forests (i.e. forest with less species per plot than in other biomes, on average), this effect is even a key one. This is actually what is illustrated by the Ref15 cited by the authors, although (as or even more) relevant studies on this issue are:

Bohn, F. J., and A. Huth. 2017. The importance of forest structure to biodiversity–productivity relationships. *Royal Society Open Science* 4.

Maréchaux, I., and J. Chave. 2017. An individual-based forest model to jointly simulate carbon and tree diversity in Amazonia: description and applications. *Ecological Monographs* 87:632–664.

Morin, X., L. Fahse, M. Scherer-Lorenzen, and H. Bugmann. 2011. Tree species richness promotes productivity in temperate forests through strong complementarity between species. *Ecology Letters* 14:1211–9.

It is also key to know whether the species present in the plots with only one species are still present in the plots with n species. Otherwise, there is a bias and the monospecific stands cannot be fairly compared with the stands with more than 1 species.

Any effect of species composition on the relationship between diversity and abundance seems quite unfeasible for several reasons: First, it is quite obvious that each of the 23 forest regions would have quite unique species pools, and therefore comparing the effect of species composition on their diversity-abundance relationship does not make sense at this scale of analysis. In fact, we might find similar species assemblages in different forest regions in terms of species richness and relative abundance, but different species composition. This has been termed the “biogeographical null hypothesis” (Cayuela et al. 2015), meaning that forest structure can be similar even though species composition is not. This indeed underlies the concept of forest biomes (ecosystems where species that have undergone similar adaptations to their surrounding environments, and therefore have similar functional traits, though their species composition are different simply because the evolutive history is different). Second, the inclusion of forest composition as a predictor in the analyses conducted within each forest region would require the inclusion of multivariate statistical approaches, which, to our knowledge, have not been developed to date in the context of SEM. We agree with the reviewer that a myriad of diversity dimensions can have a role in explaining the diversity-abundance relationship from a pure facilitative or complementarity hypothesis. However, the other way around is not neither direct nor intuitive since more individuals constitute the demographic basis to find more species, but not necessarily more functional types or phylogenetic clades. For this reason, we still consider that species richness is the most appropriate diversity variable to be consider in this theoretical frame of analysis, at least as a first step forward in this research line.

Despite these limitations when incorporating the effect of species composition to the study of diversity-abundance relationships at a global scale, we have included a small paragraph to discuss to what extent functional and phylogenetic composition might influence our results, but also how complicated it could be to integrate this dimension of diversity in such a global spatial context (L117; L127).

Cayuela, L., Gotelli, N. J., & Colwell, R. K. (2015). Ecological and biogeographic null hypotheses for comparing rarefaction curves. *Ecological Monographs*, 85(3), 437-455.

Another major concern: the management issue is central in this kind of study, as management strongly influences species composition and tree abundance in forests. I found the justification about plot selection should be strengthened (see my comment above). Pristine forests do not exist any more in many places, eg. Spain or France. The precise methodology for plot selection is missing. At least the authors should have provided an averaged number of years without management for the plots of each forest region.

We are not talking about pristine but natural forests not subjected to systematic management for long enough periods (80-100 years), so evidence of man disturbance is always exceptionally low. The difference has to do with the degree of anthropogenic influences on species composition and structure. In Europe, as well as in other parts of the world, most forests are or have been subjected to anthropogenic influences throughout their recent history. For this reason, we only considered data within protected areas and after expert knowledge consultation to assure the natural character of the study sites. In these areas, forests have been preserved from periodical thinning regimes and silviculture at least over the last 80-100 years. Eventually some of these areas may suffer small-scale human disturbances, such as occasional logging of individual trees, but these are by far a lesser relevance compared to natural disturbances such as fire, treefalls, windstorm or landslides. One can thus reasonably assume that the possibility of occasional logging would have a negligible effect on the ecological mechanisms driving species assembly and community structure.

About SEMs, the AIC comparison is not the state-of-the-art of model comparison. Are the same conclusions arise when using other metrics, for instance Chi^2 , RMSEA or BIC, which are more recommended in SEM. At least these other metrics should be provided in a supplementary material. See for instance:
Li-Chung Lin, Po-Hsien Huang & Li-Jen Weng. 2017. Selecting Path Models in SEM:

A Comparison of Model Selection Criteria. *Structural Equation Modeling: A Multidisciplinary Journal*, 24: 855-869. doi.org/10.1080/10705511.2017.1363652

Thank you for giving us the opportunity to explain this in a more extensive way. The reviewer is right in that AIC is not a good criterion in SEM statistics. We completely agree and have to say in this regard that the AIC used here is not the likelihood-based index but an adaptation of Fisher's C, as explained in Lefcheck (2015). This AIC has also been named CIC (*sensu* Cardon et al 2011) and was adjusted for small sample sizes following Lefcheck (2015). We have, nonetheless, conducted a sensitivity analysis to test whether the information criterion that we used could change the pattern obtained along the climatological NPP gradient (FAO's index). As shown in the revision, both information criteria are highly correlated ($r = 0.98$) and so using either one or the other will yield similar results (see Supplementary Material Appendix S2). Following the reviewer's advice, we finally used the Δ BIC, as it is commonly used in SEM statistics.

Cardon, M., Loot, G., Grenouillet, G., Blanchet, S. (2011). Host characteristics and environmental factors differentially drive the burden and pathogenicity of an ectoparasite: a multilevel causal analysis. *Journal of Animal Ecology*, 80(3), 657-667.

Lefcheck, J. S. (2016). *piecewiseSEM: Piecewise structural equation modelling in r for ecology, evolution, and systematics. Methods in Ecology and Evolution*, 7(5), 573-579.

I have also a major concern regarding the environmental data used. In Fig. 2, why is elevation the only environmental predictor? In most temperate and boreal regions, climatic and soil conditions will not only vary with elevation, but also across space. And as far as it is described, the plots used for each region are spread all over the "region" (often a country). Moreover, incorporating latitude or longitude is not enough, true climatic variables should be used, especially when global climate datasets at high resolution are freely available (e.g. <http://chelsa-climate.org/>).

As commented to the subject editor, we have run the analyses considering a new set of environmental variables which are integrative of climate (NPP index based on climate and energy available, NASA), soil fertility (Cation-exchange capacity), elevation (either linear and quadratic forms), and the NDVI index (at an annual resolution). We used such integrative climatic, soil or productivity variables as they have a clear biological/ecological meaning and are useful to reduce as much as possible the number of parameters to be estimated in the models, given the relatively reduced number of replicates in some of the datasets considered. The sensitivity analysis (Appendix S4)

evidences that all these predictors result in highly correlated patterns. We now report the results using elevation in the main text, as it is a commonly used proxy for environmental variability (see for instance Iverson and Prasad 1998), and also included the results obtained using other environmental variables in the supplementary material.

Iverson, L. R., & Prasad, A. M. (1998). Predicting abundance of 80 tree species following climate change in the eastern United States. Ecological Monographs, 68(4), 465-485.

In the same vein, I have doubts about the NPP values used in Fig. 3 (ref #35). Why not having used more recent proxies, eg. derived from NDVI data, again freely available at global scale?

We included NDVI in a separate SEM analysis and show that, although a similar negative pattern emerges as for NPP, the pattern is noisier. We included this result in the Supplementary Material (Appendix S5).

Other concerns

Several references were inaccurate in my opinion. For instance:

- Ref #1 does not deal with trees

This paper represents a general framing of the potential role of facilitation on plant communities. It holds on the existing literature on BEF experimental and observational evidence, as most of this evidence is supported by short-living plant communities. Nonetheless, the manuscript aims at generating a general knowledge applicable to entire ecosystems as stated by the sentence “*We demonstrate how increased environmental severity, abundance of specialist pathogens, and biological nitrogen fixation rates likely drive increased facilitation and, thus, the strength of the BEF relationship, across ecosystems*”. That is why we considered this reference a general reference in support to the idea that facilitation is a major mechanism underlying the Biodiversity-Ecosystem functioning paradigm, in plant communities in general, and in forests in particular (as they are plant communities as well).

- Ref #9 deals with deforestation, which is different than the effect of species loss in the context of the sentence (l. 28-30)

Sorry for the confusion with this sentence. We have eliminated it in order to straight up the main idea at the beginning of the introduction.

- Ref #13 is not strictly about boreal forests

Not strictly, but boreal forests represent a central question in the paper next to temperate forests. Moreover, the authors analysed temperate and boreal forests separately and so this assertion is fully supported by this reference.

- About Ref #15 & 16 (aimed at justifying space filling, eg. root segregation and canopy packing, although these terms are not mentioned): mentioning the relevance of canopy packing in ecosystem functioning with outputs from a model is a bit weak, a better reference could have been Jucker et al. (2015). Ref#16 is a bit controversial, as many works did not find any evidence for root segregation effects, eg. Meinen et al. 2009

Jucker, T., O. Bouriaud, and D. A. Coomes. 2015. Crown plasticity enables trees to optimize canopy packing in mixed-species forests. *Functional Ecology* 29.

Meinen C. et al. 2009. No evidence of spatial root system segregation and elevated fine root biomass in multi-species temperate broad-leaved forests. *Trees*. 23:941-950

Thanks. This sentence was deleted.

A relevant reference to introduce the topic could have been Waide et al. (1999):
Waide, R. B., M. R. Willig, C. F. Steiner, G. G. Mittelbach, L. Gough, S. I. Dodson, G. P. Juday, and R. Parmenter. 1999. The relationship between productivity and species richness. *Annual Review of Ecology and Systematics* 30:257–300.

Thank you very much for the suggestion. We included this reference in the revised version.

Important information about the SEM results are missing to properly assess them. The authors should provide the coefficient values for each of their arrows, for each forest region and hypotheses tested, in supplementary material for instance.

In the Supp Mat Appendix S1 we have included a table with all parameter estimates for every SEM. Besides, we report the ΔBIC between the hypotheses (abundance-richness relationship) and the null model (no relationship) in Table 1 to give statistical support to the species richness-abundance relationship in every SEM.

Self-thinning in forests is not a linear relationship. Therefore the link between tree size and abundance cannot be linear. See for instance:

Sonja Vospernik, Hubert Sterba. Do competition-density rule and self-thinning rule agree? *Annals of Forest Science*, Springer Verlag/EDP Sciences, 2015, 72 (3), pp.379-390. [ff10.1007/s13595-014-0433-x](https://doi.org/10.1007/s13595-014-0433-x). [ffhal-01284181](https://hal.archives-ouvertes.fr/hal-01284181)f

We completely agree with the reviewer that the size-abundance relationship commonly takes a negative exponential form. For this reason, a log transformation is needed for the dependent variable to linearize the expected non-linear relationship. For both abundance and richness, we used log-normal models to deal with these nonlinearities and the fact that neither richness nor abundance can take negative values. Indeed, we first applied generalized linear mixed models with a log-link function to linearize non-linear relationships. The *piecewiseSEM* package was recently updated and the *sem.fit* function has been replaced with the *psem* function. Unfortunately, many of the generalized linear models did not converge now and so we decided to apply log-normal linear models instead. Interestingly, the results using generalized and general linear models looks almost identical and so we trust that using log-normal models can be a perfect option.

Moreover, average tree size should be calculated with a quadratic mean, not arithmetic mean.

See https://www.fs.fed.us/pnw/olympia/silv/publications/opt/436_CurtisMarshall2000.pdf

Regarding the use of the quadratic mean it is important to note that this index is usually computed at the time to frame measurements of Diameter at Breast Height within a single tree, in order to minimize bias related to species-specific trunk forms. Thus, the quadratic mean represents a less biased measurement of DBH when considering different tree species. We completely agree with the reviewer on this point but have to say that this is out of our competence because all the information is given to us after technical procedures. We trust that foresters doing fieldworks and data gathering followed the standard protocols regarding this and other relevant issues. Regarding our analyses, it does not matter which of the means is used because both will return similar patterns as they are strongly correlated when used at the plot level (which is our case). In addition, most of our data sets have been previously used, tested, and published.

L. 93-94 : this is not really new, and I am surprised that the authors do not discuss the evidence for such a pattern shown in former studies (eg. Paquette & Messier 2011, cited in the study).

Ok, we now introduce a short paragraph (L114-129) that reads: *'Previous findings in north America support the idea that opposing causal paths between species richness and productivity are plausible, and that the intensity of such influences varies among biogeographical regions⁷. Our formulation, nonetheless, explicitly introduces abundance as a major component in this theoretical framework, thus allowing for the inclusion of the other way around between tree stocks and species richness. Noteworthy, the hypothesis of our paper relies on the notion that species richness (# species) can efficiently summarize species diversity at the regional level. Thus, future investigation should strive to disentangle the roles of diversity dimensions other than species richness alone, namely functional and phylogenetic diversity. Such an approach would, however, represents a major challenge because it would neither be easy nor intuitive to define how more individuals can determine species combinations or simply affect functional or phylogenetic diversity. Further efforts should thus be paid to these crossroads as a potential way towards a new paradigm of thinking when it comes to the meaning of diversity-abundance relationships in natural communities and on how they can help refine the most classical biodiversity-ecosystem functioning paradigm.'*

Reference number 7 is Paquette and Messier (2010).

I. 109-111 : Again this is not a really novel finding (see. Paquette a Messier 2011, Jactel et al. 2018) who all show how climate controls the diversity-ecosystem functioning in forests), and we all agree that the Liang et al. study is partly biased because of an oversimplification of the pattern.

Jactel, H., E. S. Gritti, L. Drössler, D. I. Forrester, W. L. Mason, X. Morin, H. Pretzsch, and B. Castagneyrol. 2018. Positive biodiversity –productivity relationships in forests: climate matters. *Biology Letters* 14.

Notice that abundance is not a measure of ecosystem functioning but rather a structural feature of communities that can be, otherwise, critical to understand the potential mechanisms underpinning the BEF relationship. This is to say that we are not talking about the biodiversity-ecosystem functioning relationship strictly, but about the diversity-abundance relationship which is inherent to the first one. Even though our findings represent an interesting element that can feed into the BEF debate, they are novel in the sense that no one has previously dealt with the abundance-diversity relationship as a critical step between biodiversity and ecosystem functioning relationships. To our knowledge this introduces a novel perspective on the topic.

The authors mention that the plot data used follow the same kind of methodology across regions. This is not true. For instance in Europe, the French NFI data do not rely on permanent plots, while the Swiss and Spanish one do. Furthermore, regarding the French NFI data, the authors do not mention how they dealt with the « simplified trees » present in the basic inventory. Indeed, a weight in these data should be calculated according to the size of the concentric plot in the French protocol (6m, 9m, 15m for small, medium, and large trees). This leads to weights of 14.14, 39.29, and 88.42 depending on the size of the tree. However, because some plots are in the border of the forest (therefore not having full size), or because of other subtleties in the protocol, the real weights of the trees differ from these values for 17% of them. And even more important: 30% of the plots contain trees for which the weight is wrong, and therefore the dendrometric values of 30% of the plot are wrong. Therefore, the weights of the trees dataset need to be corrected using the French data, and this is not mentioned at all in the ms.

Please note that the errors mentioned for French NFI plots concerned only the data as integrated within the FUNDIV project (mistake from an operator during the integration process) and more specifically the file FunDivEUROPE_all_trees_75mm (<http://fundiv.befdata.biow.uni-leipzig.de/projects/31>) (<http://fundiv.befdata.biow.uni-leipzig.de/projects/31>). It does not concern at all the raw data as provided by the French NFI available at <https://inventaire-forestier.ign.fr/spip.php?article532>. Please also note that all the values needed for our analysis (DBH, species, weight) are available for simplified trees. As a reminder, only a few variables (e.g. total height or radial growth) were not measured in simplified trees. In this study, French data were directly taken from the raw data of the NFI. Therefore, it was possible to calculate the variables reported by hectare applying the corrected weight of each individual tree (and thereafter to the specific area of the plot accordingly):

· Abundance (number of tree by hectare) = \sum individual tree weight

· Mean DBH = \sum weighttree \times DBHtree / ntree

· Richness = nb of different species on the plot

Table S1 : Species richness if known to vary with plot size. How this has been taken into account?

At the within-region level, plot size is constant, so it cannot be considered a predictor variable of richness in this section of the analyses. At the among-regions level, plot size is uncorrelated to mean richness both in the original (R

= 0.23, *P*-value = 0.28) or the harmonized (R = 0.22, *P*-value = 0.31) datasets. Nonetheless, and following your advice, we have included plot size in the last SEM model as a predictor of species richness (see Figure 4). It can be noticed that plot size does neither affect directly nor indirectly (through control on richness) the Δ BIC.

Reviewer #2 (Remarks to the Author):

This manuscript conducts interesting analyses and obtains intriguing results. The authors propose a framework to unify latitudinal gradients in tree diversity and productivity with general BEF theory by incorporating abundance-driven effects. They find diversity effects prevailing in productive (low latitude) forests only, while in less productive forests abundance-driven effects overrule possible diversity effects.

I think the analytical approach (i.e. the core results shown in Figure 3) is novel and smart. If the results hold to further scrutiny, this manuscript has potential to become an important contribution to the field.

Nevertheless, I have several conceptual concerns and a few methodological questions.

Thank you very much for the positive comments on our manuscript. We have revised the manuscript thoroughly following your advices and addressed every comment and suggestion in the main text.

Major concerns:

(1) In my opinion it is very naïve to assume that you can account for environmental variation by including elevation as a surrogate. No doubt, elevation is correlated with many environmental variables, but for a global analysis I highly doubt that elevation effects are consistent among regions and climates. What is missing are biologically relevant environmental variables related to productivity. Soil parameters come to my mind and would be appropriate, but there are likely other possibilities. As it stands, your analysis is missing an important dimension of tree growth (and thus productivity): soil fertility. It could easily be that your results are at least partly driven by differences in soil characteristics, considering that many of the highly weathered tropical soils are actually rather poor in nutrients, which might at least partly explain your latitudinal results. You should either add an analysis showcasing that elevation does, as you claim, account for environmental variation within and among forests regions, or add relevant additional data. To keep your current analytical framework (which I like) one could condense

environmental variables to one or two principal components and use them as predictors in the path models.

Thank you very much for this comment. Following your advice, we have re-analyzed the data using a set of predictor variables including elevation in linear and quadratic forms, but also a 1-km resolution NPP index, a 1-km resolution NDVI index (as suggested by reviewer #1), a 250 -m resolution Cation-exchange Capacity index (CEC, as a surrogate of soil fertility) and the combination of NPP and CEC. We include the new analyses in the supplementary material (Appendix S4) and show how all the results obtained strongly point to the same pattern. In other words, results using the different predictors support the previous findings that diversity effects on abundance prevail towards low latitudes, in highly productive forest biomes of the world (Figure S3 and Table S3, Supp Mat Appendix S4).

(2) I know you want to sell your paper with global warming, but it is unclear to me how this feeds back into your analysis. Generally, I think you are overselling a lot. Your basic ecological findings are interesting enough. No need to construct an utter link to global warming, also when considering that temperature is just one of several components of NPP.

Thank you, we have rephrased the text in some parts to avoid overselling of the climate change idea. Nonetheless, we still believe that our findings can be discussed and applied in the frame of the nature-based mitigations strategies to face climate change. For this reason, we maintain part of the original paragraphs.

(3) The text is unclear in parts. Many concepts are mixed or not used in the appropriate context. This might potentially limit the accessibility of the manuscript for a general readership. I will point out a few problematic passages in the line-by-line comments below. For example, the 'diversity effects' hypothesis named by you is in the literature usually referred to as 'biodiversity-ecosystem functioning' or short 'BEF'.

We do agree with the idea that BEF is the main context for the discussion in this paper but have to say that the biodiversity-ecosystem functioning hypothesis is a somewhat larger theoretical framework than the abundance-diversity relationship strictly. Specifically, the BEF does consider functioning as a major component of the hypothesis. For this reason, and to be conservative, we prefer to designate our hypothesis just as a diversity effects hypothesis. We do not have any inconvenience on using BEF, but still consider that using BEF could lead to confusion among the readership of NCOMM.

(4) Actually, a part of your core result has been demonstrated before. Dormann et al. (<https://doi.org/10.1101/524363>) reanalyzed the dataset of Liang et al. (your reference 3), finding that the claimed general and positive relationship between tree diversity and productivity only holds for few ecoregions. This matches well with your findings.

Thank you very much for the reference. We agree that Dormann's findings proof that BEF are not that spread nor global. They also found that, for some biomes, BEF is not statistically significant, but still lack in stablishing a potential mechanistic hypothesis in support to the lack of a global BEF in forest ecosystems. Their results are clearly in line what we propose here. Again, we are not directly talking about the BEF but address the abundance-diversity relationship, which is a major component of the first one, but not the same. We have included this literature in our manuscript and discuss it as to highlight the significance of our results in the frame of the state of the art in BEF research (L61).

Methodological questions:

(5) How were the data selected. Was there some sort of data filtering? This is currently obscure but very important information.

We included only data on well-conserved forest regions in which human disturbance was absent or unsystematic and residual. Accordingly, we used data corresponding to forest sites located in protected areas and natural reserves. We created a specific Appendix in the Supplementary Material to ensure that the selection criteria used is clear enough (Appendix S7).

(6) How did you account for the differing forest inventory methods among forest regions? Whether you measure DBH for individuals exceeding 5 cm or 10 cm matters a lot for stem counts (i.e. abundance). This might have a large impact on the outcome of tests of the 'more individuals hypothesis'.

To check for this reasonable query, we harmonized the whole dataset by considering only trees larger than 10 cm DBH in every regional dataset. We included this in the supp mat as a sensitivity analysis. As you will see (Supp Mat Appendix S3), results obtained using the original and the harmonized data strongly support the same pattern without any statistical difference in the regression analysis (Supp Mat Figure S2). We also included in the last SEM analysis the potential artifact of plot size on richness and thus with the potential to affect the NPP- Δ BIC relationship. Results show that plot size is unrelated to plot species richness directly, or to Δ BIC indirectly.

(7) As you are probably aware, the formulation of AICc depends on sample size (i.e. number of plots in your case). A difference in 2 AICc units means something different if you apply it to a dataset with 40 replicates or if you apply it to a dataset with 400 replicates. In the less replicated data, 2 units mean a relatively larger difference in model fit compared to the dataset with tenfold more replicates. According to your Table S1, number of plots varied by more than one order of magnitude. You should account for this in your analyses, if you want to compare AICc difference among forest regions. At least you have to demonstrate that the relationship in Figure 3 is not systematically influenced by differences in plot numbers. In this regard, just referring to the correlation coefficients in Figure S1 is not sufficient.

Thank you for giving us the opportunity to clarify that the AICc used in the previous version is not the classical AIC (likelihood-based index) but the CIC (sensu Cardon et al 2011) as used by Lefcheck (2016) in the piecewiseSEM package. This index is computed based on the Fisher's C statistic as suggested by Shipley (2013). Nonetheless, and following the advice of reviewer #1, we used the BIC index, in which penalization of n allows to support the most parsimonious models independently of n. We have included a comparison between results obtained using BIC and AIC(CIC) in the Supp Mat Appendix S2. As can be seen, both results are strongly correlated (r=0.99).

Shipley, B. (2013). The AIC model selection method applied to path analytic models compared using ad-separation test. Ecology, 94(3), 560-564.

Minor comments (line-by-line):

L 4: Packaging: This is too much jargon for the opening sentence.

Ok, replaced by 'abundance'

L 6: See the Dormann et al. preprint referred to above for an opposing view.

Ok, included.

L 9: You critically omit soil fertility from your considerations.

We did so because soil fertility is not explicitly considered in the frame of the 'more individuals' hypothesis. In fact, productivity in the frame of the MIH is mostly associated to available energy, which indeed is the cause for abundance in natural populations. Said that, we completely agree with you that soil fertility is a major component of fertility in the broad sense and therefore should be accounted for.

L 19ff: I can not see how your findings have the claimed 'major impacts' on climate change mitigations. Generally, you are overselling your results. The conclusions here are extremely vague and can not directly be deduced from your findings.

We have rephrased the text as to make it clearer and more tied to our aim at linking this to climate change mitigation strategies.

L 28: It is often claimed that forests are losing tree species. Yet, this is mostly due to forestry or maybe invasive species. As you have worked in unmanaged forests only, I am not sure that this strong statement is justified here.

We have eliminated this part of the first paragraph and focused on tree species diversity only. Thank you for this comment.

L 34-35: What you frame as 'diversity effects' hypothesis has long been named as 'biodiversity-ecosystem functioning'.

We understand this comment suggested that the "more species" and "biodiversity-ecosystem functioning" hypotheses are tightly related. However, the BEF includes explicitly the functioning of ecosystems, which represents a major step beyond the goal of our paper. The richness-abundance relationship is only a part of the whole BEF hypothesis, as we did not deal explicitly with ecosystem functions in our question, but a static picture of structure. Of course, the causality on the richness-abundance relationship fully entails with the BEF discussion and raises a new perspective to add in the general debate. That is why we, in an attempt to be conservative, decided to talk about 'more species' hypothesis and not BEF.

L 40: It might be important to add that growth-trait inference depends on tree diversity, with usually stronger influences of traits on growth when tree diversity is larger. See the recent paper by Bongers et al (2020, Journal of Ecology 108: 256-266).

Thank you very much for this suggestion. We have included it in the main text almost literally.

L 44: Be aware that the formulation of the 'more individuals hypothesis' uses energy as the primary currency. The problem I see here is that relative productivity (if we take it as surrogate for energy) changes in trees when trees grow (i.e. when abundances get lower but productivity at individual level gets larger). This might or

might not be a violation of the principal assumptions of the MIH and it would be good if you develop this thought further.

Thank you very much for the comment. We completely agree in that more energy (whenever combined with more water) can result in increased individual growth. However, this individual growth increment, even if it also controls for abundance following the Yoda's law paradigm, does not necessarily deny more individuals compared with sites in the same ontogenetic stage under less available energy. For this reason, and assuming that mean tree size must be included in the model to control for the Yoda's law, there is not necessarily contradiction with the MIH, because it can be assumed that, at a similar ontogenetic stage, more energy determines more individuals. Accordingly, we included mean tree size as a community-level variable that summarizes for the negative trend in local abundance associated to forest development and shelf thinning. We thus see that both the Yoda's law and the MIH are compatible.

L 61: How were the 21 regions/datasets selected? This is important to know.

We selected forests in protected areas and pristine lands such as the Amazonia or the Himalayas. We have created a new Appendix in the Supplementary Material (Appendix S7) for a more in-depth explanation of forest data selection. Following these criteria, we have included two more datasets available in France and Spain as suggested by the editor. Nonetheless, we have been very conservative in this process what, along with the reduced accessibility to forest data at the region level, implies important restrictions to data selection.

L 72ff: This is a really smart analysis.

Thank you very much

L 99: Delete 'very strong'. In general, you could tune down on the valuing language.

Ok, thank you. Deleted

L 120: What is 'proper'? Unclear.

We have deleted proper for clarity

L 120ff: Again, I think this is too much overselling in regards to climate change.

Although we understand your point, we think that this derivative is really relevant, especially under the global crisis we are facing and the necessity to design new protocols for conservation of species diversity in natural forest ecosystems. In any case, we have shortened this paragraph to tune down the climate change mitigation message (paragraph in L128-137).

L 121: The multifunctionality statement can also be made for productive sites. See e.g. Schuldt et al. (2018, Nature Communications 9: 2989) for a study from tropical forests.

We agree with the reviewer that the role of tree diversity on multifunctionality can also be applied to productive forests. Nonetheless, it is hard to understand how species richness and abundance cause one each other when human control both through management and silviculture. We didn't include this in the reference since we are dealing with natural forests only and its inclusion will require an specific explanation, so enlarging the main text and the main objectives.

L 138: 'comparable protocols and clear sampling design'. This is totally vague. We need much more detail in a supplementary file.

We included literature at the end of Table S1 if readers are interested in any dataset. We also created Appendix S7 for further explanation of data selection and brief description of sampling protocols.

L 168: As explained above, I question whether elevation is sufficient for capturing between plot variation.

We have included (in the Supplementary Material) a sensitivity analysis using other environmental variables namely, NPP, NDVI, elevation, soil Cation Exchange Capacity (CEC) and also the combination of NPP and CEC. Results reflect a similar pattern of higher probability for a diversity effects hypothesis towards low latitudes, mostly rainy tropical forests, regardless of the variables used (Supp Mat Figure S3 and Table S3).

L 174: Please be more specific. For what models did you add random effects. What was the random effect structure? Random intercept or random slope models?

We included a new paragraph (L189-200) in the Methods section to explain in detail the structure of random terms and the models.

Figure 3: Please add the fitted mean to the plot.

Ok, done

Table S1: (1) Please unify the number of digits you give. For example, three digits for elevation measurements make no sense at all. Similar issues prevail for almost all columns. Make sure the displayed number of digits match to the accuracy of measurements.

Ok, done

Table S1: (2) Column 'NPP_median': Isn't it strange that BR and US(NY) have exactly the same NPP values? Same problem for other pairs (e.g. US(SE) vs. RU).

Yes, that was a typo. Thanks for noticing. We have the correct values in the revised version of the table

Table S1 (3) How were richness values intrapolated?

Yes, we initially interpolated richness values assuming a linear relationship. However in the revised version we use the average value of species richness at plot level.

Reviewer #3 (Remarks to the Author):

The study aims to look at the relationship between richness and abundance in forest plot data, and to determine the casual direction of this relationship using SEMs. I think the question (what is the causality of the relationship) is interesting, and the dataset is impressive. However, there are several shortcomings that need to be addressed.

Main Points

One of the main issues is that the English needs a lot of improvement. This is not just a pedantic complaint, but I really struggled to understand a lot of what was written, particularly in the methods, which means it is hard to know exactly what was done. Some examples:

Lines 173 -175 – more information is needed about what this test is for and how it is implemented

Ok. We have included an extensive paragraph to explain this more in detail (L169-189) :"*To explore which of the two competing hypotheses is supported in each of the twenty-three forest regions considered, we analysed the direction of the richness-abundance relationship by applying Structural Equation Models (SEM) following the theoretical framework defined in Fig. 2. The two hypotheses differ in*

the causal direction of the richness-abundance relationship (direction of the arrow linking species richness and abundance). The two candidate models share three equations: (1) abundance as a linear function of mean tree size and elevation, (2) richness as a linear function of elevation, and (3) mean tree size as a function of elevation. On the contrary, the SEM testing the 'more species' hypothesis (MSH) included (4) abundance as a function of richness, whereas the SEM testing the 'more individuals' hypothesis (MIH) included (5) richness as a function of abundance. Elevation was included as a surrogate of climate variation within each forest region to account for the influence of environmental variability on diversity, mean tree size and abundance. We tested the roles of environmental variables other than altitude in a sensitivity analysis, namely NPP, Normalized Difference Vegetation Index (NDVI), Cation Exchange Capacity (CEC) and the combination of NPP and CEC. Results of these analyses can be found in the supplementary material (Appendix S4). Both richness and abundance per plot, as discrete variables, were log-transformed to meet the normal distribution of residuals and homoscedasticity. Besides, the log-transformation of abundance allowed for the linearization of the relationship abundance-mean tree size, which is known to adopt negative exponential forms²⁴. We applied linear models in the regression analyses using the function lm of the package stats in R³⁵"

203 – 206: I am not sure what this analysis is doing?

Ok. We have explained this in detail (L222-230): "We modelled the ΔBIC as a function of the climatological NPP but also included species richness as a potential confounding effect of latitude, and plot size differences among regions to control for the species richness variability as an artefact associated with the different sampling protocols regionally. We thus evaluated the role of species richness as a potential direct predictor of the ΔBIC in the presence of NPP. In doing so, we can evaluate whether the observed pattern in ΔBIC is just a consequence of a confounding relationship between species richness and NPP along the latitudinal gradient or determined by a net climatological influence."

207 – 208: what does trusted missing paths mean?

We have rephrased the text for clarity (L221-236): "To analyse whether climate controls the prevalence of one hypothesis over the other at the global scale we used a SEM with piecewiseSEM package in R. We modelled the ΔBIC as a function of the climatological NPP but also included species richness as a potential confounding effect of latitude, and plot size differences among regions to control for the species richness variability as an artefact associated with the different sampling protocols regionally. We thus evaluated the role of species richness as a potential direct predictor of the ΔBIC in the presence of NPP. In doing so, we can evaluate whether the observed pattern in ΔBIC is just a consequence of a confounding relationship between species richness and NPP along the latitudinal gradient or determined by a

net climatological influence. As a sensitivity analysis for the use of NPP as a global-scale environmental variable we used NDVI (1km resolution) instead of the climatological NPP. Results suggested that, even if the pattern holds for NDVI consistently with NPP, the climatological NPP was a better candidate for the model ($BIC_{NPP} = 41.87$, $BIC_{NDVI} = 63.73$; Fig. S5). Additionally, we tested the support for the missing paths (latitude on ΔBIC) in the SEM using Fisher's C statistic. Specifically, missing paths will be negligible when the P-value associated to the Fishers C statistic is larger than 0.05"

There are other predictors that should be used in the SEMs. Plot size for one – I think this is a major variable to not be kept constant. It is well established that richness increases with area. Why not include plot size in the SEMs as an additional co-variate driving richness? You could also use mixed effects SEMs (which you briefly mention) and include plot inventory and/or region as random effects. I think these also need to be accounted for.

Thank you very much for the advice. As you will see, Fig 4 now has a SEM that includes variable plot size as a control of species richness. Please note that this SEM was defined to analyse the relationship between NPP and ΔBIC , and so there is not replicability within regions. For this reason, a random term does not make sense in this step because there is no pseudo-replication at the regional level.

The fact that the minimum threshold of trees included varies between plots could also affect results. Why not run a sensitivity test and filter out all trees below 10cm (i.e. the highest minimum threshold across studies) DBH in all plots and re-run the analyses?

Thank you very much for this comment as well. As mentioned above we have conducted a sensitivity analysis to test whether results coming from the original data are different from harmonized data (minimum tree size included 10 cm DBH). As you can see in Appendix S3, both analyses support a similar pattern. Moreover, we considered a new sensitivity analysis including both data to test whether the effect 'type of data' influence the response of ΔBIC to Latitude. The result of this analysis supported the idea that results are not affected for the type of data (See Supp. Mat. Appendix S3): "To formally evaluate whether original or harmonized data introduce any discrepancy on the model outputs regarding the latitudinal patterns in ΔBIC , we fitted a linear model in which the ΔBIC is expressed as linear function of the interactive effects of latitude and the type of data (original/harmonized). This interaction implies that latitudinal patterns in ΔBIC varies depending on the type of data considered (i.e. the slope of the relationship change significantly between the original and harmonized data). Results using the Akaike Information Criterion corrected for small sample sizes (AIC_c , Hurvich and Tsai 1989) rejected such interactive effects ($AIC_{interaction} = 306.9$; $AIC_{no-interaction} = 305.5$). Moreover, and even if influences of latitude on ΔBIC are assumed to remain

constant, the ΔBIC could be comparatively higher or lower depending on the type of data (i.e. the main effects associated to the type of data influence the intercept in the linear model). To unveil this, we evaluated whether the type of data has a significant contribution to the ΔBIC using a linear model. The results using the $AICc$ showed negligible effects of the type of data on the ΔBIC ($AICc_{with\ type\ of\ data} = 305.5$; $AICc_{without\ type\ of\ data} = 305.0$)."

What happens with the goodness of fit test if Fisher's C is significant? Is the model excluded? Would it not make more sense to exclude it before comparing $AICc$? So as a first step test the goodness of fit of your two competing models to determine if they are acceptable, and if both pass this step, then compare?

We firstly compared models using the BIC before we tested whether the best supported model is relevant or not compared to a null model in which the abundance-richness relationship is not considered. If we would check for the Fisher's C firstly, we would have focused on the missing paths and not on causality in the abundance-richness relationship. Thus, the relevance of missing paths and their potential consequences should be evaluated later, and only once the causal direction is set according to the BIC (or CIC) criterion.

It is not clear why you need to use elevation as a proxy and not climate variables themselves?

Thanks for this comment. We have updated the analyses including other potential environmental variables such as NPP, NDVI, elevation (linear and quadratic forms) and Cation-exchange capacity (as a surrogate of soil fertility). We decided to run models including one or two such variables at a time to test whether the observed pattern changes or not. We found that the same results are achieved irrespective of the predictor variable considered. We include these results as a sensitivity analysis in Supp Mat Appendix S4.

Other Points

I think it needs to be clearer early on how gaining a better understanding of causality in the relationship will help climate change mitigation. You start by talking about biodiversity crises and ecosystem functioning etc, but the link between these big issues and this particular relationship is not clear. In the final paragraphs at the end of the paper it becomes clearer, but perhaps stress this early on as well. An example from early on in the paper: you state "Maintaining and enriching tree assemblages could thus help mitigating climate change via more dense packaging

and efficient resource use". But, based on what is written before this sentence, the link here is not clear.

Thank you very much for this interesting comment. We have rephrased the introduction to 1) reduce the climatic change context (as suggested by reviewer 2#) and 2) to clarify links between climate change and the importance of tree species richness in natural forests.

Lines 43 – 46: Could be clearer that the More Individuals Hypothesis states that a greater amount of energy means more individuals which means more species. That first part of this (i.e. the energy part) is unclear in your description

Absolutely, thank you very much. We clarified this in L(36-40) : *"By contrast, higher available energy can promote species richness and tree biomass storage by promoting abundance¹⁵. This idea, framed as the 'more individuals' hypothesis, assumes that the number of species is solely a probabilistic product of abundance in the sense that viability of natural populations is contingent on the number of available individuals¹⁶."*

48 – 49: But it has also been found not to hold in a large number of cases. The Storch et al paper you cite also distinguishes between the strict formulation and the weaker formulation of the hypothesis, which I think is worth mentioning. And they argue based on their theory that the strict formulation does not hold, but this is not mentioned.

Thank you for this suggestion. We have introduced the flaws and weaknesses of the hypotheses, which also offered us the opportunity to reinforce the idea that both hypotheses are plausible and that it is thus necessary to disentangle whether and where one prevails over the other (L43-48): *"Nonetheless, it seems that only the soft formulation of the hypothesis is plausible since species richness can indeed explain abundance patterns in some cases¹⁹. Any attribution of richness or abundance as the cause and/or the consequence of one another therefore remains a challenge. Yet, defining this causal relation is a prerequisite to understand the likely mechanisms underlying the correlational evidence on the richness-biomass storage relationship available in forest ecosystems."*

Is diversity – abundance actually an accurate description? As diversity metrics usually incorporate a measure of abundance. Perhaps richness – abundance relationship is more accurate.

We agree with the reviewer that some diversity indices incorporate abundance as a component to control for equitability. We thus have replaced the diversity-abundance relationship description with richness-abundance

relationship for clarity.

The wording around the use of AICc needs changing throughout the text. For example, lines 74-75, you mean AICc was used to compare models to test the prevalence ... And line 85, say something like "Our results indicate". 99-100: I am not sure what delta AICc supporting diversity effect means?

Ok, we rephrase these sentences as to accurately refer to the information criteria (L94-99): *"To formally evaluate under which conditions one of the contrasting causal hypotheses prevails over the other at the global scale, we analysed the relation between the ΔBIC (indicative of prevailing diversity effects when negative and more individuals hypothesis when positive) and the FAO's (region averaged) climatological net primary productivity (NPP) index using a Structural Equation Model (SEM) (similar results using NDVI instead of NPP were yielded as shown in Appendix S5)."*

Line 100 – add the coefficient for NPP as well as the R2

Ok, done

Reviewers' Comments:

Reviewer #2:

Remarks to the Author:

Dear authors,

I am generally happy with the way you have dealt with my criticism. You have either explained your point or performed the suggested changes.

Some further line-by-line comments:

L44: What is a 'soft' formulation?

L122: This is colloquial language (at least to me). Please revise.

L123: Replace 'paper' by 'study'.

L127: 'represent' not 'represents'.

The author list of reference 8 appears incomplete. Kühn is definitely not the senior author.

Appendix 5 (L306): You probably mean 'ns' and not 'ms'. In general, I found rather a lot of typos and further inconsistencies in the supplementary material.

Reviewer #3:

Remarks to the Author:

I reviewed the paper previously and it is definitely now improved and better written, and the authors have dealt with most of my previous comments. However, I have a few further points for consideration.

The first sentence of the abstract is confusing as it defines the more species hypothesis without any reference to abundance, meaning that it isn't clear why the MSH and the MIH, using these definitions, are contrasting hypotheses. More broadly, could there be a situation where both hypotheses are in fact true, to varying degrees? A sort of positive feedback? For example, more energy supports more individuals which allows for more species (to be sampled from the regional pool) which through niche partitioning means then even more individuals can persist?

The main analyses use elevation, as you say, as a proxy for climate variables. And then you say you use climate variables as a sensitivity test. This seems strange as why not just use that as the main analysis, as by doing the sensitivity test with climate variables you are showing you don't need a proxy?

I think you should outline why you choose climatological NPP as your climate variable of choice, i.e. why is it better than using say just raw climate variables (e.g. mean temperature). I think you should provide a definition of what 'climatological NPP' is when you first mention it in the paper and methods. There is a description in one of the figure headings, but it could be better explained and justified in the text itself.

In table S1, what are the area units for plot size? And can you also include the range in plot richness, if this is the mean. And the number of individuals. Lots of the mean richness values are very low (e.g. 1 or 2), could this be an issue? Can you expect to find much effect of richness on abundance if the richness is 1 in most plots? Knowing the range in richness would make this clearer.

Line 102 – Isn't delta BIC just a measure of the relative support of both hypotheses, not "supporting the more species hypothesis"?

Lines 196 – 198 – what were these lms used for? Do you mean what was used inside the SEMs?

Elevation and altitude are different things so be consistent with what you are using.

I think the title should read World's main forest biomes rather than main World's

REVIEWER COMMENTS

Reviewer #2 (Remarks to the Author):

Dear authors,

I am generally happy with the way you have dealt with my criticism. You have either explained your point or performed the suggested changes.

Thank you very much for the constructive review and the positive comments.

Some further line-by-line comments:

L44: What is a 'soft' formulation?

Storch et al (2018) in their paper in Ecology Letters refer to the soft formulation of the MIH as the hypothesis in which environmental stochasticity or colonization and speciation rates can, along with abundance, control the number of species. On the contrary, the strict formulation suggests that species richness is explained by energy through abundance only. To clarify this term, in the newer version of the manuscript we have included the following statement in brackets: "(which considers environmental stochasticity besides energy)". (L 45-46)

L122: This is colloquial language (at least to me). Please revise. **OK, paragraph revised**

L123: Replace 'paper' by 'study'. **OK**

L127: 'represent' not 'represents'. **OK**

The author list of reference 8 appears incomplete. Kühn is definitely not the senior author. **OK, we have corrected the reference**

Appendix 5 (L306): You probably mean 'ns' and not 'ms'. In general, I found rather a lot of typos and further inconsistencies in the supplementary material. **OK, we have reviewed the supplementary material thoroughly.**

Reviewer #3 (Remarks to the Author):

I reviewed the paper previously and it is definitely now improved and better written, and the authors have dealt with most of my previous comments. However, I have a few further points for consideration.

Thank you very much for the positive comments. Below, you can find responses to every comment and question addressed.

The first sentence of the abstract is confusing as it defines the more species hypothesis

without any reference to abundance, meaning that it isn't clear why the MSH and the MIH, using these definitions, are contrasting hypotheses.

We agree in that the first sentence was slightly confusing without an explicit mention to the concept of 'abundance'. We now have incorporated this in the revised version as follows: *"More tree species can increase the carbon storage capacity of forests (here referred to as the 'more species' hypothesis) through increased tree productivity and tree abundance due to complementarity, but also be the consequence of increased tree abundance through increased available energy ('more individuals' hypothesis)".* (L4-8)

More broadly, could there be a situation where both hypotheses are in fact true, to varying degrees? A sort of positive feedback? For example, more energy supports more individuals which allows for more species (to be sampled from the regional pool) which through niche partitioning means then even more individuals can persist?

Absolutely. We agree with the reviewer on this comment and thank the advice on the positive feedback (cool). Throughout the text, we always stress that one hypothesis prevails over the other in order to be consistent with the idea that even if one of them is more likely this does not deny the existence of the other one. That is why we stated in the revised version the following sentence as to be clear that MIH mechanisms do not imply necessary the absence of MSH mechanisms: *"Therefore, for as long as temperature, precipitation, or the combination of both will limit net primary productivity, mechanisms underpinning the 'more species' hypothesis (such as complementarity) will have less influence than the climatic filtering has on functional strategies and tree abundance."* (L 109-111).

Following the advice of the reviewer we have included almost literally his/her idea: *"Moreover, a positive feedback between both types of mechanisms might be plausible in the sense that in environmental contexts where more energy supports more individuals, and this allows for more species (to be sampled from the regional pool), increased niche partitioning will mean then that even more individuals can persist. Future research should be focus on unveiling such potential positive feedbacks between more energy and complementarity driving ecosystem functioning in forests and other ecosystems worldwide."* (L 112-118).

The main analyses use elevation, as you say, as a proxy for climate variables. And then you say you use climate variables as a sensitivity test. This seems strange as why not

just use that as the main analysis, as by doing the sensitivity test with climate variables you are showing you don't need a proxy?

Thank you very much for giving us the opportunity to clarify this issue. We understand the concern related to elevation and, for this reason, we need to stress why do we choose elevation as the variable reported in the main text. On the one hand, we prefer maintaining elevation as the environmental proxy since this is, indeed, the environmental variable measured in the raw dataset. Elevation has been largely recognized a good composite variable of temperature, and precipitation variability, but also climatic extremes (Pepin et al., 2015; Revadekar et al 2013), atmospheric pressure and edaphic conditions (Ali et al 2019) in forests. On the other hand, elevation has been commonly used as an integrative environmental proxy explaining gradients of species richness, tree density and forest dynamics at local to regional scales (among the many papers on this topic see for instance Zhu et al 2019, Malizia et al 2020, Homeier et al 2010, Benavides et al 2016). Importantly, we do not have any other environmental variable measured in the field, so all we can use instead are interpolated variables such as the NPP, and CEC, both derived from interpolated precipitation, temperature, elevation and, in the case of CEC, lithological conditions. If we bear in mind that elevation is used by climate reanalyses model for spatial interpolation, then it is rather reasonable to consider that elevation is the most reliable and accurate variable for summarizing the most important environmental variability at the regional level in large-scale studies, such as the one presented here. In fact, even if we would dispose of a set of field information on precipitation, temperature, soil and geomorphological factors, we should look for any variable (PCA components for instance) to reduce noisy information and the number of parameters to be estimated, giving the limiting number of samples in some of the forest datasets included in our study. Our experience says that elevation would be highly related to the first axes of these multivariate techniques.

For all these reasons we still consider that elevation is the best candidate variable to be presented in the main text as an environmental proxy that affect species richness, abundance and mean tree size at the regional scale. Otherwise, the readers can easily access to the supplementary material to see that results using the different environmental proxies obtained from climatic and digital elevation models (NPP, CEC, NDVI) are almost equal. We have included in the main text the following paragraph: *"Elevation has been largely recognized a good composite variable of temperature, and precipitation variability, but also climatic extremes³⁶, atmospheric pressure and edaphic conditions³⁷ in forests. On the other hand, elevation has been commonly used as an integrative environmental*

proxy explaining gradients of species richness, tree density and forest dynamics in local to regional spatial contexts^{38,39,40,41}. We tested the roles of environmental variables other than elevation in a sensitivity analysis, namely Elevation (second order polynomial), climatological Net Primary Productivity (NPP, Miami model), Cation Exchange Capacity (CEC) and Normalized Difference Vegetation Index (NDVI). Results of these analyses can be found in the supplementary material (Appendix S4).” (L.203-211)

References

- Ali, S., Hussain, I., Hussain, S., Hussain, A., Ali, H., & Ali, M. (2019). Effect of Altitude on Forest Soil Properties at Northern Karakoram. *Eurasian Soil Science*, 52(10), 1159-1169.
- Benavides, R., Escudero, A., Coll, L., Ferrandis, P., Ogaya, R., Gouriveau, F., ... & Valladares, F. (2016). Recruitment patterns of four tree species along elevation gradients in Mediterranean mountains: Not only climate matters. *Forest Ecology and Management*, 360, 287-296.
- Homeier, J., Breckle, S. W., Günter, S., Rollenbeck, R. T., & Leuschner, C. (2010). Tree diversity, forest structure and productivity along altitudinal and topographical gradients in a species-rich Ecuadorian montane rain forest. *Biotropica*, 42(2), 140-148.
- Malizia, A., Blundo, C., Carilla, J., Osinaga Acosta, O., Cuesta, F., Duque, A., ... & Calderón-Loor, M. (2020). Elevation and latitude drives structure and tree species composition in Andean forests: Results from a large-scale plot network. *PLoS one*, 15(4), e0231553.
- Pepin, N., Bradley, R. S., Diaz, H. F., Baraër, M., Caceres, E. B., Forsythe, N., ... & Miller, J. R. (2015). Elevation-dependent warming in mountain regions of the world. *Nature climate change*, 5(5), 424-430.
- Revadekar, J. V., Hameed, S., Collins, D., Manton, M., Sheikh, M., Borgaonkar, H. P., ... & Baidya, S. (2013). Impact of altitude and latitude on changes in temperature extremes over South Asia during 1971–2000. *International Journal of Climatology*, 33(1), 199-209.
- Zhu, Z. X., Nizamani, M. M., Sahu, S. K., Kunasingam, A., & Wang, H. F. (2019). Tree abundance, richness, and phylogenetic diversity along an elevation gradient in the tropical forest of Diaoluo Mountain in Hainan, China. *Acta Oecologica*, 101, 103481.

I think you should outline why you choose climatological NPP as your climate variable of choice, i.e. why is it better than using say just raw climate variables (e.g. mean temperature). I think you should provide a definition of what ‘climatological NPP’ is when you first mention it in the paper and methods. There is a description in one of the figure headings, but it could be better explained and justified in the text itself.

It is important to note that we do not have raw climatological data but interpolated climatic data at spatial extents ranging from 1 x 1 km to 10 x 10 km. The problem of using precipitation and temperature data is related to the overparameterization of the model, since productivity implies non-linear

interactive effects of precipitation and temperature which are complex to be managed in linear models. The NPP index based on the Miami model (Lieth 1975) is a well calibrated variable based on non-linear interactive effects of temperature and precipitation according to the following equation:

$$NPP_T = 3000(1 + \exp(1.315 - 0.119 \times T))^{-1}$$

$$NPP_P = 3000(1 - \exp(-0.000664 \times P))$$

$$NPP = \min(NPP_T, NPP_P)$$

Where NPP_T and NPP_P represent the components of primary productivity associated to temperature and precipitation, respectively. Thus, the NPP results from the combination of both components (more details in Lieth 1975). We have included a new paragraph in the methods section to clarify why the NPP is a good climatic proxy in our study and how it is derived from the previous equations following the Miami model: *“This NPP index is assessed as a non-linear combination of temperature and precipitation following the equations in the Miami model³⁵:*

$$NPP_T = 3000(1 + \exp(1.315 - 0.119 \times T))^{-1}$$

$$NPP_P = 3000(1 - \exp(-0.000664 \times P))$$

$$NPP = \min(NPP_T, NPP_P)$$

Where NPP_T and NPP_P represent the components of primary productivity associated to temperature and precipitation, respectively. NPP will increase with rising temperature and precipitation up to a saturation point of 3000 gDM/m²/year (DM stands for dry matter). Interestingly, this index assumes that both temperature and precipitation will limit net primary productivity.” (L.178-188)

Reference

Lieth, H. 1975. Modelling the primary productivity of the world. Pages 237–264 in H. Lieth and R. H. Whittaker, editors. Primary productivity of the biosphere. Springer-Verlag, New York, New York, USA.

In table S1, what are the area units for plot size? And can you also include the range in plot richness, if this is the mean. And the number of individuals. Lots of the mean richness values are very low (e.g. 1 or 2), could this be an issue? Can you expect to find much effect of richness on abundance if the richness is 1 in most plots? Knowing the range in richness would make this clearer.

Following the advice of Ref3# we have incorporated to table S1 1) the units to plot size (m²), the species richness ranges, and a new column with plot abundance info (means and ranges).

Line 102 – Isn't delta BIC just a measure of the relative support of both hypotheses, not "supporting the more species hypothesis"?

We have replaced 'supporting the more species hypothesis' with '(relative support to both hypotheses)' (L103).

Lines 196 – 198 – what were these lms used for? Do you mean what was used inside the SEMs?

Yes. We have clarified this in the revised version: "*Inside the SEM models, we applied regression analyses using the function lm of the package stats in R35.*" (L215-217)

Elevation and altitude are different things so be consistent with what you are using.

We have reviewed this as to be consistent throughout the text.

I think the title should read World's main forest biomes rather than main World's
OK, we have changed the title accordingly.